# ADAPTING LARGE LANGUAGE MODELS VIA READING COMPREHENSION

**Daixuan Cheng** [‡], **Shaohan Huang**[*†] **& Furu Wei** [†]
[†] Microsoft Research   [‡] Beijing Institute for General Artificial Intelligence (BIGAI)
https://huggingface.co/AdaptLLM

## ABSTRACT

We explore how continued pre-training on domain-specific corpora influences large language models, revealing that training on the raw corpora endows the model with domain knowledge, but drastically hurts its prompting ability for question answering. Taken inspiration from human learning via reading comprehension—practice after reading improves the ability to answer questions based on the learned knowledge—we propose a simple method for transforming raw corpora into reading comprehension texts. Each raw text is enriched with a series of tasks related to its content. Our method, highly scalable and applicable to any pre-training corpora, consistently enhances performance across various tasks in three different domains: biomedicine, finance, and law. Notably, our 7B language model achieves competitive performance with domain-specific models of much larger scales, such as BloombergGPT-50B. Furthermore, we demonstrate that domain-specific reading comprehension texts can improve the model's performance even on general benchmarks, showing the potential to develop a general model across even more domains. Our model, code, and data are available at https://github.com/microsoft/LMOps.

## 1 INTRODUCTION

The proliferation of general large language models (LLMs) has given rise to the emergence of domain-specific large language models. Existing methods can be broadly classified into three approaches. The first trains models from scratch on a mixture of domain-specific and general corpora (Wu et al., 2023b). While this intuitively creates domain-specific LLMs, the substantial computational and data requirements raise significant concerns (Yang et al., 2023; Ling et al., 2023). The second fine-tunes the language model using supervised datasets (Singhal et al., 2022; 2023; Li et al., 2023b;a; Wang et al., 2023; Han et al., 2023; Xiong et al., 2023; Huang et al., 2023), offering a more cost-effective option. However, there are uncertainties about how well fine-tuned LLMs grasp domain knowledge that can be universally applied to all domain-specific tasks, as discussed by Zhou et al. (2023) and Gudibande et al. (2023). The third prompts the general language model with retrieved domain knowledge (Li et al., 2023b; Cui et al., 2023; Huang et al., 2023), which can be considered as an application of LLM rather than a direct enhancement to the LLM itself.

Continued pre-training on domain-specific corpora, also known as domain-adaptive pre-training (Gururangan et al., 2020), has been proven effective in adapting various natural language understanding models (Devlin et al., 2019; Liu et al., 2019; Clark et al., 2020) to specific domains (Yao et al., 2021; Gururangan et al., 2020; Cheng et al., 2022). This approach enables language models to leverage general ability while incorporating domain-specific knowledge, benefiting downstream domain-specific tasks at reduced costs. This motivates our investigation into whether domain-adaptive pre-training also benefits large-scale generative models. We conduct preliminary experiments on three domains—biomedicine, finance, and law—revealing that continued training on the raw corpora results in a drastic drop in prompting performance but still benefits fine-tuning and knowledge probing evaluations. This leads us to conclude that domain-adaptive pre-training using raw corpora endows the LLM with domain knowledge while hurting its prompting ability.

To leverage domain-specific knowledge while enhancing prompting performance, we introduce a simple method for transforming large-scale raw corpora into reading comprehension texts: each raw

---

[*]Corresponding author: shaohanh@microsoft.com

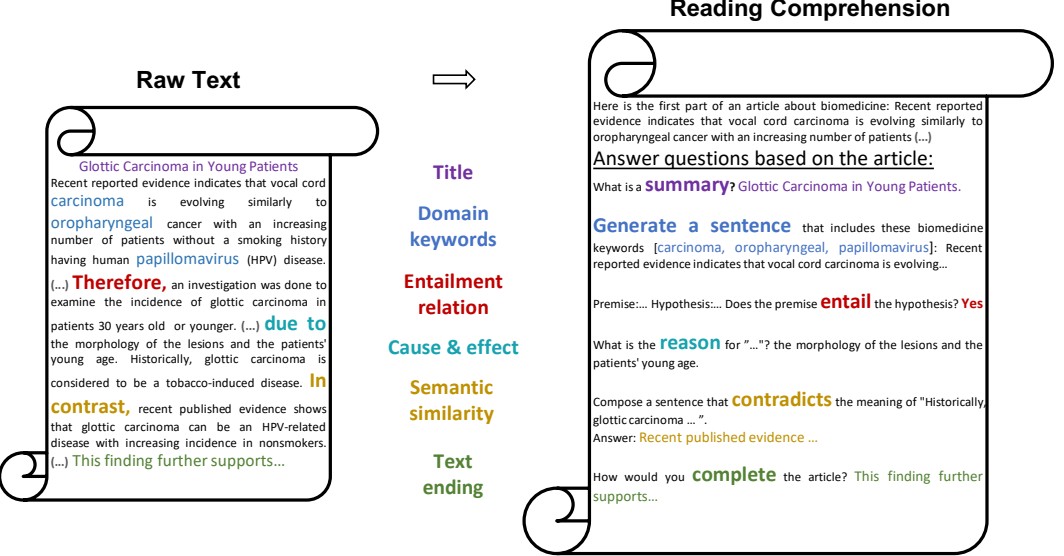

Figure 1: **A simplified example of a reading comprehension text**, wherein the raw text is followed by a series of tasks constructed from it, including Summarization (purple), Word-to-Text (blue), Natural Language Inference (red), Commonsense Reasoning (teal), Paraphrase Detection (yellow), and Text Completion (green). The complete version is in Appendix G.

text is enriched with a series of tasks relevant to its content, as illustrated in Figure 1. These tasks are designed to help the model maintain its ability to answer questions using natural language, based on the context of the raw text. Furthermore, we augment the reading comprehension texts with diverse general instructions, thereby further enhancing prompting ability (Wei et al., 2022; Zhou et al., 2023; Xu et al., 2023; Mukherjee et al., 2023). Our experiments in domains such as biomedicine, finance, and law highlight the effectiveness of our approach in improving model performance on various domain-specific tasks. We refer to this resulting model as AdaptLLM, for Adapted Large Language Model. Looking ahead, we envision extending this methodology to the development of a general large language model, contributing to the ever-expanding landscape of tasks across more domains.

In summary, our contributions include:

- We investigate domain-adaptive pre-training for large language models, where we find continued training on domain-specific raw corpora can endow the model with domain knowledge, but drastically hurts its prompting ability.
- We propose a simple recipe which automatically converts large-scale raw corpora into reading comprehension texts, to effectively learn the domain knowledge while concurrently preserving prompting performance.
- Our experiments show the effectiveness of our method in consistently improving model performance in three different domains: biomedicine, finance and law.

## 2 PRELIMINARY EXPLORATION ON DOMAIN-ADAPTIVE PRE-TRAINING

Given the proven efficacy and efficiency of domain-adaptive pre-training in adapting natural language understanding models (Gururangan et al., 2020; Yao et al., 2021; Cheng et al., 2022), we embark on an exploration to ascertain whether this method remains effective for large-scale generative models. We continue to train the general LLaMA (Touvron et al., 2023a) on the domain-specific raw corpora of biomedicine, finance, and law, respectively, and conduct prompting, fine-tuning, and knowledge probing evaluations to assess the model performance within each domain (detailed experiment settings are in Section 4).

**Prompting vs. Fine-tuning.** In Table 1, when conducting fine-tuning evaluation on the domain-specific tasks, the model that has undergone domain-adaptive pre-training consistently outperforms

Table 1: **Domain-specific task performance** of the general language model (General LLM) and the model that has undergone vanilla domain-adaptive pretraining (DAPT (Gururangan et al., 2020)). We report the average of task scores within each domain under prompting, fine-tuning and knowledge probing settings.

| Method | Prompting | | | Fine-tuning | | | Knowledge Prob | |
|---|---|---|---|---|---|---|---|---|
| | BioMed. | Finance | Law | BioMed. | Finance | Law | BioMed. | Law |
| General LLM | **44.2** | **58.6** | 34.2 | 64.2 | 79.9 | 42.0 | 36.5 | 45.0 |
| DAPT | 41.7 | 57.6 | **35.0** | **66.5** | **80.9** | **45.4** | **36.9** | **45.6** |

the general model across all three domains. This aligns with findings about language understanding models (Gururangan et al., 2020), indicating that continued pre-training enriches the language model with domain-specific knowledge. Paradoxically, a contradictory trend emerges in the prompting performance, where a noticeable drop is observed across most domains after domain-adaptive pre-training. This contradiction leads us to hypothesize that while vanilla domain-adaptive pre-training enhances the LLM's domain knowledge, contributing to the fine-tuning improvements, it also significantly impairs its ability to perform well in prompting, causing the observed drop.

**Domain Knowledge Probing.** To confirm whether the language model gains domain knowledge from domain-adaptive pre-training, we employ a probing method similar to LAMA (Petroni et al., 2019). Using the supervised datasets available in each domain as the basis, we create domain-specific knowledge-probing datasets. The dataset creation process is detailed in Appendix A. In Table 1, we present the results of domain knowledge probing for the biomedicine and law domains[1]. In both domains, we observe improved results after domain-adaptive pre-training, indicating that the model indeed acquires domain knowledge.

The analyses above suggest that the drop in domain-specific prompting performance can be attributed to the reduced prompting ability. This reduction may result from the limited diversity of the pre-training corpora within a specific domain (Longpre et al., 2023b), which limits the diversity of input-output patterns derived from raw texts (Wei et al., 2022). Therefore, improving prompting ability becomes essential for effectively harnessing the domain knowledge acquired from domain-adaptive pre-training.

## 3 ADAPTING LARGE LANGUAGE MODELS VIA READING COMPREHENSION

Instead of continuing to train the large language model on domain-specific raw corpora, we convert the raw corpora into reading comprehension texts and adapt the model using the converted data. In reading comprehension, each raw text is followed by a series of tasks related to its content. We regard the model training phase on the raw text as the "reading" phase, and the subsequent training on the relevant tasks as the "comprehension" phase (Chen et al., 2023; Gu et al., 2022; 2023). These comprehension tasks follow the question-answering format, aimed at enriching the model's prompting ability to respond to input questions (Wei et al., 2022). This design is inspired from human learning, where practice after reading enhances the ability to answer questions based on the acquired knowledge. Besides, we augment the training data with general instructions (Zhou et al., 2023; Xu et al., 2023; Mukherjee et al., 2023) to benefit from the diversity of input-output patterns, thereby further improving prompting ability.

### 3.1 CREATING READING COMPREHENSION TEXTS

To create reading comprehension texts, we start by mining intrinsic tasks from the raw corpora with a handful of mining patterns. The idea of mining tasks from pre-training corpora was introduced by van de Kar et al. (2022). This method mines intrinsic tasks through a few regex-based patterns, and then fine-tunes the model on these tasks to enhance zero-shot performance. Our approach

---

[1]We were unable to construct a knowledge probing test for finance due to the limited availability of supervised datasets in this domain.

leverages the self-supervised nature of this mining strategy. This enables us to scale up the transfer of raw pre-training corpora, capitalizing on the domain-specific knowledge embedded in the raw texts and the enhanced prompting ability provided by the comprehension tasks.

Table 2: **Mining patterns and input-output templates.** For mining, {VERBAL} is replaced with the verbalizers in Table 3, {WORD} captures a single word, and {SENT} captures a single sentence. Each input-output template is paraphrased into multiple variations. We also turn the task around—exchanging the input and output—to enhance task diversity.

| Task Type | Mining Pattern | Input-output Template |
|---|---|---|
| *Summarization* | | |
| Title | Title as summary | `What is a summary?` {TITLE} |
| Topic | {SENT1} {VERBAL} {SENT2} | {SENT1} `is about:` {SENT2} |
| *Word-to-Text* | | |
| Word-to-text | Domain keywords as input; sentence as output | `Generate a sentence about these` {DOMAIN} `keywords` [{WORD1}, {WORD2}, {WORD3}]: {SENT} |
| Definition | {WORD} {VERBAL} {SENT} | `How to define` {WORD}? {SENT} |
| *Natural Language Inference* | | |
| Entail Neutral Contradict | {SENT1} {VERBAL}, {SENT2} | `Does "`{SENT1}`" entail "`{SENT2}`"?` {Yes/Maybe/No} |
| *Commonsense Reasoning* | | |
| Cause-effect | {SENT1} {VERBAL}, {SENT2} | `What is the` {effect/cause} |
| Effect-cause | {SENT1} {VERBAL} {SENT2} | `of` {SENT1}? {SENT2} |
| *Paragraph Detection* | | |
| Similar Different | {SENT1} {VERBAL}, {SENT2} | `Compose a sentence to` {support/ contradict} `"`{SENT1}`".` {SENT2} |
| *Text Completion* | | |
| Text completion | Text ending as completion | `How would you complete the article?` {ENDING} |

Table 2 gives an overview of the techniques used to mine and create tasks from each raw text. Phrases like `Answer questions based on the article:` are employed to concatenate the raw text with the followed tasks, as illustrated in Figure 1. Additionally, we paraphrase each input-output template to multiple variations and turn the task around to enhance task diversity (Wei et al., 2022; Chung et al., 2022; Longpre et al., 2023a).

**Summarization** prompts the models to generate a concise summary of the provided text, encouraging them to extract its main idea. We use queries such as `What is a summary?` to prompt the model to summarize the raw text, with the text title serving as the groundtruth. We also reverse the task, asking the model to craft an article based on the given title.

Additionally, we prompt the models to identify sentence topics. To unearth such intrinsic tasks, we utilize regex-based patterns to identify sentences aligning with the patterns specified in Table 2. We then employ the corresponding templates to construct the input-output pairs (van de Kar et al., 2022).

**Word-to-Text** enhances the model's grasp of domain-specific vocabulary by prompting it to generate sentences incorporating specific words. To identify domain-specific words, we use the SentencePiece tool (Kudo & Richardson, 2018) to build a vocabulary from the target domain corpora. We then compare this domain vocabulary to the general language model's vocabulary, considering

Table 3: **Verbalizers for mining patterns in Table 2.**

| Task Type | Verbalizer |
|---|---|
| *Summarization* | |
| Topic | `talks about`, `is about`, `'s topic is` |
| *Word-to-Text* | |
| Definition | `is defined as`, `'s definition is` |
| *Natural Language Inference* | |
| Entail | `Yes`, `Therefore`, `Thus`, `Accordingly`, `Hence`, `For this reason` |
| Neutral | `Maybe`, `Furthermore`, `Additionally`, `Moreover`, `In addition` |
| Contradict | `No`, `However`, `But`, `On the contrary`, `In contrast`, `Whereas` |
| *Commonsense Reasoning* | |
| Cause-effect | `Therefore`, `Thus`, `Accordingly`, `Hence`, `For this reason` |
| Effect-cause | `due to`, `on account of`, `owing to` |
| *Paragraph Detection* | |
| Similar | `Similarly`, `Equally`, `In other words`, `Namely`, `That is to say` |
| Different | `No`, `However`, `But`, `On the contrary`, `In contrast`, `Whereas` |

words present in the domain vocabulary but absent from the general vocabulary as domain-specific. Next, we filter out tokens with fewer than 10 characters, resulting in a set of domain-specific keywords.

For each sentence in the raw text, we count the number of domain-specific keywords. Sentences having more than three domain-specific keywords are selected for making Word-to-Text tasks. We take the domain-specific keywords in the sentence as the input, asking the model to generate a sentence with `Generate a sentence that includes these {DOMAIN} keywords`.

We also turn the task around by taking the sentence as input and asking the model to find the keywords about the target domain, using `What keywords about {DOMAIN} can be extracted from this sentence?`. Here we point out the target domain by replacing {DOMAIN} with domain names such as `biomedicine`, `finance`, or `law`. Besides, we prompt the model to define concepts using the mining patterns and input-output templates in Table 2.

**Natural Language Inference** concerns how two sentences relate, typically asking, given a first sentence, whether a second sentence is true, false or possibly true. We use the regex-based patterns in Table 2 to search for "premise-hypothesis-relation" triplets within the raw text. For example, we categorize the relation between two sentences as "Entailment" if they are connected by the verbalizer `Therefore`, and as "Contradictory" if connected by `However`.

Additionally, we enhance diversity by reformatting classification tasks into generation tasks. For instance, when the relation between two sentences is "Entailment", we employ templates like {SENT1} `Thus?` to prompt the model to generate an output, where the ground truth is the second sentence.

**Commonsense Reasoning** evaluates the ability to perform physical or scientific reasoning while considering common sense. We identify cause-and-effect logic within sentences using the regex-based patterns in Table 2. We then formulate the input-output pairs using templates such as `What is the reason of {SENT1}? {SENT2}`.

**Paraphrase Detection** asks a model to determine whether two sentences are semantically equivalent. To collect task data, we use regex-based patterns in Table 2 to search for "sentence1-sentence2-label" data triplets. However, we empirically find that the mining patterns cannot consistently identify two sentences with strictly equivalent meanings. For instance, sentences linked by the verbalizer `Similarly` may not share similar meanings.

Therefore, we reformat the classification task into a generation task to reduce dependence on label accuracy. Instead of inquiring whether two sentences are similar, we prompt the model to generate a

sentence that either supports or contradicts the meaning of a given sentence, using input-output templates like `Can you create a sentence that contradicts the meaning of {SENT1}? {SENT2}` when the mined label is "Different".

**Text Completion.** In addition to the inherent casual language modeling task within generative language models, we insert queries such as `How would you complete the article?` between sentences to prompt the language model to generate the subsequent section. An advantage of Text Completion task is that it does not require any specific mining patterns, thus can be applied to any raw texts.

## 3.2 MIXING WITH GENERAL INSTRUCTIONS

While we have designed diverse mining patterns, input-output templates and task reversals to enhance prompting ability, they might not fully address the infinite task diversity in real-world scenarios. In light of this, we propose to mix the reading comprehension texts with general instructions to cover a wider range of input-output patterns.

## 4 EXPERIMENT SETTINGS

**Domain-adaptive Pre-training.** PubMed Abstracts and FreeLaw Opinions from the Pile (Gao et al., 2021) are utilized as the pre-training corpora for the biomedicine and law domains, respectively. For finance, we collect financial news from May 2022 to May 2023[2] for over $7,000$ stocks, using the FinGPT codebase (Yang et al., 2023). General instructions are sourced from LIMA (Zhou et al., 2023), WizardLM (Xu et al., 2023), and Orca (Mukherjee et al., 2023; Lian et al., 2023). We continue to train LLaMA-7B (Touvron et al., 2023a) on each domain, and explore different ratios for mixing reading comprehension texts with general instructions; the optimal ratios for biomedicine, finance, and law are $1:1$, $1:2$, and $1:1$, respectively. Implementation details, dataset specifications, and other pre-training hyper-parameters can be found in Appendix B.

**Creating Reading Comprehension Texts.** Using the mining patterns in Table 2, we search for sub-categories within each task type. To prevent task dominance, we limit the number of task examples per sub-category to two for each raw text. For each mined example, we randomly sample from various paraphrased or task-reversed templates to generate an input-output example. To structure the reading comprehension text, we use $\backslash n\backslash n$ to connect comprehension tasks and link them with the raw text. On average, about two input-output examples are collected per reading comprehension text. Please refer to Appendix C for mining pattern implementation details and Appendix G for cases of reading comprehension texts.

**Domain-specific Tasks.** For biomedicine, we evaluate on PubMedQA (Jin et al., 2019), ChemProt (Kringelum et al., 2016), MQP (McCreery et al., 2020), RCT (Dernoncourt & Lee, 2017), and USMLE (Jin et al., 2020). For finance, we evaluate on the five publicly available tasks also evaluated by BloombergGPT (Wu et al., 2023b): ConvFinQA (Chen et al., 2022), FPB (Malo et al., 2014), FiQA SA (Maia et al., 2018), Headline (Sinha & Khandait, 2020), and NER (Alvarado et al., 2015), and adopt similar prompting settings with BloombergGPT. For law, we evaluate on SCOTUS (Spaeth et al., 2020), CaseHOLD (Zheng et al., 2021) and UNFAIR-ToS (Lippi et al., 2019) from the LexGLUE (Chalkidis et al., 2022) benchmark. Evaluation details are provided in Appendix D.

## 5 MAIN RESULTS

In Table 4, we compare our models (AdaptLLM) with the general language models, and the models that have undergone vanilla domain-adaptive pre-training on raw corpora (DAPT). On various tasks in the three domains, the use of raw corpora in DAPT adversely affects the prompting performance. However, the transformation of raw corpora and the inclusion of general instructions in AdaptLLM manage to counteract this effect, outperforming the general language model.

**Biomedicine.** We compare with MedAlpaca (Han et al., 2023), which fine-tunes LLaMA (Touvron et al., 2023a) on medical question-answering instructions. While supervised instructions help

---

[2]Access to earlier news is limited.

Table 4: **Domain-specific task performance** of the general language models, the models that has undergone vanilla domain-adaptive pre-training (DAPT), and ours (AdaptLLM) in prompting evaluation. We also display prompting results of MedAlpaca (Han et al., 2023) in biomedicine, BloombergGPT (Wu et al., 2023b) in finance, and LexGPT (Lee, 2023) in law. AdaptLLM-13B in biomedicine is trained from LLaMA-13B and AdaptLLM-6B in law is trained from GPT-J-6B.

| Biomedicine | PubMedQA | ChemProt | MQP | RCT | UMSLE | AVERAGE |
|---|---|---|---|---|---|---|
| LLaMA-7B | 59.6 | 31.4 | 50.7 | 45.1 | 34.5 | 44.2 |
| DAPT-7B | 52.6 | 26.6 | 49.2 | 46.6 | 33.5 | 41.7 |
| MedAlpaca-7B | 58.6 | **39.0** | 50.7 | 40.8 | **36.7** | 45.1 |
| AdaptLLM-7B | **63.3** | 35.2 | **54.4** | **50.4** | 33.1 | **47.3** |
| LLaMA-13B | 59.6 | 42.8 | 49.3 | **56.7** | 34.7 | 48.6 |
| DAPT-13B | 51.1 | 38.0 | 49.0 | 50.9 | 34.6 | 44.7 |
| MedAlpaca-13B | 60.7 | 38.4 | 57.4 | 51.3 | **41.2** | 49.8 |
| AdaptLLM-13B | **66.0** | **47.6** | **73.0** | 50.4 | 34.0 | **54.2** |

| Finance | ConvFinQA | FPB | FiQA SA | Headline | NER | AVERAGE |
|---|---|---|---|---|---|---|
| BloombergGPT-50B | 43.4 | 51.1 | 75.1 | 82.2 | 60.8 | 62.5 |
| LLaMA-7B | 29.2 | 55.9 | 69.2 | 77.7 | **61.1** | 58.6 |
| DAPT-7B | 29.6 | 55.3 | 64.9 | 77.5 | 60.6 | 57.6 |
| AdaptLLM-7B | **41.5** | **62.5** | **72.1** | **81.4** | 59.3 | **63.4** |

| Law | SCOTUS | | CaseHOLD | | UNFAIR-ToS | AVERAGE |
|---|---|---|---|---|---|---|
| | mic-F1 | mac-F1 | mic-F1 | mac-F1 | | |
| GPT-J-6B | 15.9 | 13.6 | **34.9** | **34.9** | 79.8 | 35.9 |
| DAPT-6B | 10.1 | 10.5 | 34.6 | 34.6 | **84.9** | 35.0 |
| LexGPT-6B | 16.9 | 7.7 | 27.0 | 27.0 | 81.9 | 32.1 |
| AdaptLLM-6B | **18.8** | **20.1** | 34.7 | 34.7 | 80.0 | **37.7** |
| LLaMA-7B | 28.3 | 10.8 | 32.9 | 32.9 | 65.8 | 34.2 |
| DAPT-7B | 25.0 | 9.8 | 34.2 | 34.2 | 72.0 | 35.0 |
| AdaptLLM-7B | **30.0** | **17.8** | **35.1** | **35.1** | **74.4** | **38.5** |

MedAlpaca outperform LLaMA in some domain-specific tasks, this advantage isn't consistent, possibly because the instructions don't fully infuse domain knowledge, or the instructions specific to one domain struggle with various input-output scenarios.

**Finance.** We compare our results with those reported in BloombergGPT (Wu et al., 2023b), a model trained from scratch on a mixture of financial and general corpora. While LLaMA-7B scores lower than BloombergGPT-50B, AdaptLLM-7B achieves competitive performance with it. This highlights the computational and data efficiency of our approach compared to training from scratch.

**Law.** We compare with LexGPT (Lee, 2023) which conducts vanilla domain-adaptive pre-training on GPT-J (Wang & Komatsuzaki, 2021) using the raw corpora of Pile of Law (Henderson et al., 2022). In contrast to GPT-J, LexGPT shows negative prompting results. This trend aligns with our observation in section 2 that continued pre-training on domain-specific raw corpora leads to worse prompting performance. However, our method contributes to positive prompting results, emphasizing the effectiveness of comprehension tasks and general instructions.

## 6 ABLATIONS ON TRAINING DATA

Table 5 presents ablation results on different training data and data mixtures: (1) **Raw Text** refers to the raw corpora used in vanilla domain-adaptive pre-training. (2) **Read. Compre.** converts the raw corpora into reading comprehension texts, boosting the prompting ability to show better results

in all of the adapted domains. (3) **Gen. Ins.** refers to general instructions. (4) **Read. + Gen. Ins.** augments reading comprehension texts with general instructions. Compared to using reading comprehension texts only, the inclusion of general instructions further improves the prompting ability, leading to better task results. Moreover, compared to the use of general instructions alone, the utilization of reading comprehension texts provides domain knowledge that enhances performance in domain-specific tasks. Furthermore, we provide ablations for each of the comprehension task types in Appendix E, where we find that Word-to-Text and Natural Language Inference exhibit the highest effectiveness on domain-specific tasks.

Table 5: **Ablation results on training data.** Raw Text refers to raw corpora, Read. Compre. refers to reading comprehension texts, Gen. Ins. refers to general instructions, and Raw. + Gen. Ins. and Read. + Gen. Ins. correspond to different data mixtures. We report the average of task scores in prompting evaluation within each domain.

| Data | Raw Text | Read. Compre. | Gen. Ins. | Raw. + Gen. Ins. | Read. + Gen. Ins. |
|------|----------|---------------|-----------|------------------|-------------------|
| BioMed. | 41.7 | 44.3 | 43.3 | 44.8 | **47.3** |
| Finance | 57.6 | 60.0 | 62.2 | 61.7 | **63.4** |
| Law | 35.0 | 37.0 | 37.8 | 34.7 | **38.5** |

# 7    ANALYSIS OF DOMAIN KNOWLEDGE AND PROMPTING ABILITY

Our design of reading comprehension is to learn the domain-specific knowledge from the raw texts and to enhance the prompting ability from the comprehension tasks. In this section, we conduct analyses on the two aspects respectively.

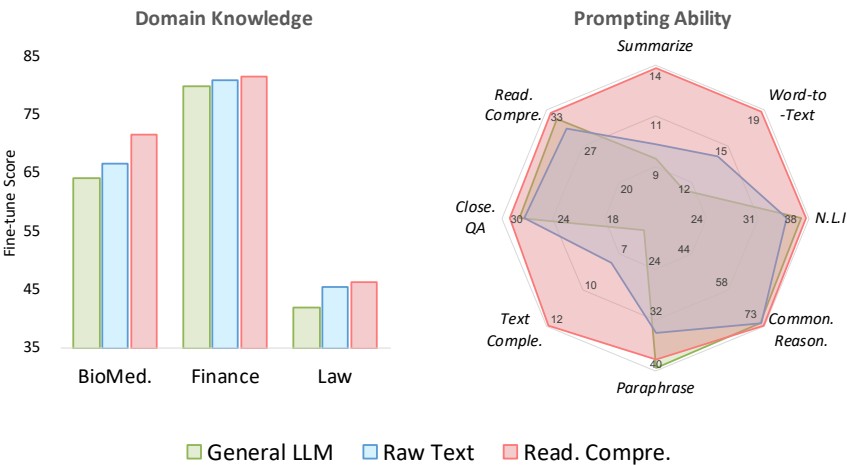

Figure 2: **Fine-tuning evaluation on domain-specific tasks (left) and prompting evaluation on general tasks (right)**. General LLM is the general language model, Raw Text trains the general model on the domain-specific raw corpora, and Read. Compre. trains the general model on the reading comprehension texts constructed based on the raw corpora. We report the average of task scores within each domain/type, detailed results are listed in Appendix F.

**Domain Knowledge.** In addition to the prompting evaluation in Sections 5 and 6, we conduct fine-tuning and knowledge probing evaluations to assess whether the continued training on reading comprehension texts endows the general model with domain knowledge. As shown in the fine-tuning results in Figure 2, after the training on reading comprehension texts, the model consistently exhibits improved results on the domain-specific tasks. The fine-tuning and knowledge probing improvements (detailed in Appendix A) provide empirical evidence that the reading comprehension texts indeed imbue the general model with domain knowledge.

Notably, Read. Compre. outperforms Raw Text in all the adapted domains in the fine-tuning results. This may be because the inclusion of diverse comprehension tasks naturally creates a "multi-task instruction tuning" setting, which benefits single-task fine-tuning (Longpre et al., 2023a).

**Prompting Ability.** Our approach focuses on enhancing prompting ability through the comprehension tasks. To assess their effectiveness, we employ general LLM benchmarks to evaluate zero-shot prompting performance. Specifically, we evaluate at least three general tasks for each comprehension task type, following the task clustering setup in FLAN (Wei et al., 2022). Besides, we assess performance on general Reading Comprehension and Closed-book QA tasks to evaluate the ability to answer questions with or without contexts.

Figure 2 presents the average task scores within each task type, subsequently averaged across the three adapted language models. Converting raw texts into reading comprehension texts consistently improves prompting performance across all task types. Remarkably, when trained on our domain-specific reading comprehension texts (without the inclusion of general instructions), we achieve even better results than the general language model on most task types. This highlights our approach's potential in developing a general language model across more domains. In Appendix E, we provide ablations for each comprehension task type to analyze its impact on related downstream tasks.

# 8 RELATED WORK

Recent works that apply large language models to specific domains (Singhal et al., 2022; 2023; Li et al., 2023b; Wu et al., 2023a; Li et al., 2023a; Wang et al., 2023; Xiong et al., 2023; Wu et al., 2023b; Yang et al., 2023; Cui et al., 2023; Huang et al., 2023) can be categorized into three main approaches: training from scratch, instruction fine-tuning and retrieval-augmented prompting.

**Training from Scratch.** Training a domain-specific language models from scratch is an intuitive approach to realize domain specialization. BloombergGPT (Wu et al., 2023b) represents an early example of large language models in the financial domain, trained on a mixture of financial and general corpora. It demonstrates great performance on financial tasks without sacrificing the performance on general LLM benchmarks. However, studies (Yang et al., 2023; Ling et al., 2023) have pointed out "training from scratch" comes with expensive computational and data requirements, which motivates the need for low-cost adaptation methods such as fine-tuning or continued pre-training.

**Instruction Fine-tuning.** Fine-tuning large language models on domain-specific tasks, particularly those involving question-answering instructions, serves as a cost-effective adaptation method (Singhal et al., 2022; 2023; Li et al., 2023b;a; Wang et al., 2023; Han et al., 2023; Xiong et al., 2023; Huang et al., 2023). However, models fine-tuned with limited data might struggle to acquire sufficient domain knowledge. Therefore, creating large-scale supervised data becomes significant objective. Previous methods employ high-performing LLMs (OpenAI, 2023) to generate question-answering pairs (Li et al., 2023a), but the inference cost can be substantial. In such situations, harnessing large-scale domain-specific corpora is a promising for acquiring domain knowledge.

**Retrieval-augmented Prompting.** Retrieval augmentation enhances LLMs by integrating external domain-specific information without modifying the model parameters (Li et al., 2023b; Cui et al., 2023; Huang et al., 2023). This enables LLMs to better answer domain-specific questions and address issues like hallucination. It's important to allow LLMs the option to accept or reject retrieved information due to potential incompleteness or conflicts (Ling et al., 2023). Training LLMs to incorporate domain knowledge can aid in making such informed acceptance or rejection decisions.

# 9 CONCLUSION

This paper focuses on adapting large language models via continued training on domain-specific corpora. We propose a simple method to transform large-scale domain-specific raw corpora into reading comprehension texts, enabling the model to acquire domain knowledge from raw texts and to enhance prompting ability through comprehension tasks. Experiments in three different domains confirm the effectiveness and generalizability of this method. Moreover, the reading comprehension texts enhance model performance on general LLM benchmarks, suggesting potential for improving general language models across more domains. We hope our method can inspire further exploration into adapting large language models with the use of large-scale pre-training corpora, efficiently empowering language models for downstream tasks in specialized areas.

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

## A    DOMAIN KNOWLEDGE PROBING

We devise domain knowledge probing tests to determine whether continued training on the domain-specific texts can enhance the model's domain-specific knowledge. Our probing test design is inspired by LAMA (Petroni et al., 2019), where the task format closely resembles the pre-training task. This allows us to analyze the model's inherent knowledge without altering its architecture (e.g., adding a model head) or parameters (e.g., fine-tuning). LAMA utilizes "fill-in-the-blank" cloze statements to match the masked language modeling task of BERT (Devlin et al., 2019). Similarly, we create "predict-the-next-token/sentence" tests to align with the casual language modeling tasks of generative language models (Radford & Narasimhan, 2018). Table 6 presents the knowledge probing results in the biomedicine and law domains. We observe that continued training on domain-specific raw/reading comprehension texts indeed endows the large language model with domain knowledge.

Table 6: **Domain knowledge probing results** of general large language model (**General LLM**), the model trained on domain-specific raw corpora (**Raw Text**) and the model trained on the reading comprehension texts constructed based on the raw corpora (**Read. Compre.**).

| Domain | General LLM | Raw Text | Read. Compre. |
|--------|-------------|----------|---------------|
| BioMed. | 36.5 | 36.9 | 36.8 |
| Law | 45.0 | 45.6 | 46.4 |

**Biomedicine.** To create a knowledge probing test for the biomedicine domain, we utilize the MedM-CQA (Pal et al., 2022) dataset. This dataset comprises numerous high-quality multiple-choice questions, covering diverse healthcare topics and 21 medical subjects. To align the testing format with casual language modeling, we remove data samples in the instruction-following format. These include samples starting with question words like "What", "Who" and "When", or ending with ":", "?", and "-". Additionally, samples having the fill-in-the-blank marker "_" are also removed.

The evaluation is similar to zero-shot prompting: we feed into the model the raw data input, without any demonstrations, and then compare per-token-likelihood of each option to get the model prediction. This evaluation is conducted individually for each of the 21 medical subjects, and the average score across all the subjects is reported.

**Law.** For the law domain knowledge probing, we employ the LEDGAR dataset (Tuggener et al., 2020). This dataset is designed for contract provision classification and encompasses a wide spectrum of 100 distinct law topics. Each label represents the principal topic of the given contract provision. Originally structured as a 100-classification task, we adapt it for knowledge probing by simplifying it into a four-choice option format. For each data sample, we preserve the label class and randomly select three additional classes to create the four candidate options.

Similar to biomedicine knowledge probing, we feed into the model the data input using the template "{CONTRACT} The topic is", without any demonstrations, and then compare per-token-likelihood of each option to get the model prediction. The evaluation is performed individually for each of the 100 law topics, and the average score across all the topics is reported.

## B    DOMAIN-ADAPTIVE PRE-TRAINING

Table 7 presents specifications of the pre-training corpora in each domain and Table 8 presents pre-training hyper-parameters. A $<pad>$ token is added to the model vocabulary for sentence padding. Our pre-training code is based on TorchScale (Ma et al., 2022)[3]. In each domain, we explore different ratios for mixing domain-specific raw/reading comprehension texts with general instructions, specifically considering ratios of $1 : 2$, $1 : 1$, and $2 : 1$. The end-of-sentence token $$ is used to concatenate between documents, where a document could be a raw text, a reading comprehension text, or a general instruction.

---

[3]https://github.com/microsoft/torchscale

Table 7: **Pre-training corpora.**

| Domain | Data Source | Raw Size | # Tokens | # Docs |
|--------|-------------|----------|----------|--------|
| BioMed. | PubMed Abstracts (Gao et al., 2021) | 19.3 GiB | 5.4 B | 15.5 M |
| Finance | Stock News | 5.1 GiB | 1.2 B | 1.1 M |
| Law | FreeLaw Opinions (Gao et al., 2021) | 51.2 GiB | 16.7 B | 3.6 M |

Table 8: **Hyper-parameters of domain-adaptive pre-training.**

| Hyper-parameter | Assignment |
|-----------------|------------|
| Computing infrastructure | 32 V100-32GB GPUs |
| Run-time | 24 Hours |
| Number of steps | 10,000 |
| Batch size | 32 |
| Maximum sequence length | 2,048 |
| Maximum learning rate | 1e-5 |
| Optimizer | Adam |
| Adam beta weights | 0.9, 0.95 |
| Learning rate scheduler | cosine |
| Weight decay | 0.1 |
| Warm-up steps | 1000 |
| Gradient clipping | 1.0 |
| Dropout ratio | 0.1 |

## C  CREATING READING COMPREHENSION TEXTS

**Title Collection for Summarization Tasks.** In the biomedicine domain, the title for each raw text in PubMed Abstracts (Gao et al., 2021) is the first sentence within the text, separated from other sentences by a newline character $\backslash n$. In the finance domain, we specifically collect the titles when gathering news using the FinGPT codebase (Yang et al., 2023). In FreeLaw Opinions corpora (Gao et al., 2021) of the law domain, there are no explicit titles for individual raw texts. Instead, titles are available for some sections within the raw text. Hence, we initially divide each raw text into sections and gather as many titles as possible from these sections. Subsequently, we consider one section, rather than an entire raw text, as the basis for creating one reading comprehension text.

**Regex Pattern Implementation.** Before mining task examples, we truncate each raw text to its initial 1,800 tokens, enabling the insertion of comprehension tasks within a maximum sequence length of 2,048. For tasks which employ regex patterns to mine task examples, we fill in the patterns with the corresponding verbalizer and identify sentences that match the patterns. This process of expanding patterns into regular expressions follows van de Kar et al. (2022): {`VERBAL`} is substituted with a capturing group that incorporates all verbalizers, separated by the alternation operator `|`. For instance, the verbalizer set `Therefore, Thus, Hence` is expanded to `(Therefore|Thus|Hence)`. Subsequently, the keywords listed in Table 9 are replaced with the corresponding regular expressions. The result is a regular expression containing capturing groups for extracting sentences.

## D  DOMAIN-SPECIFIC TASKS EVALUATION

**Prompting.** In prompting evaluation, each task corresponds to multiple prompt templates and we randomly sample one of them for each data example, to mitigate result variance caused by template sensitivity. Prompt template examples are presented in Table 11. Following Brown et al. (2020), we classify tasks into two question types to get model predictions: 1) For multiple-choice questions, we compare the per-token likelihood of each option to determine the model prediction; 2) For text completion questions, we employ greedy search to get the free-form answer.

Table 9: **Keywords that compile into regular expressions.** These keywords are used in the mining patterns and verbalizers (van de Kar et al., 2022).

| Keyword | Regex |
|---|---|
| {VERBAL} | Replaced with the verbalizer |
| {WORD} | regex: `([^.!?\n,;\"\s]{10,})`
Matches a single word having more than 9 characters |
| {SENT} | regex: `([^.!?\n]{50,}[.!?]+)`
Matches a single sentence having more than 50 characters |

Our prompting settings in the finance domain follow BloombergGPT (Wu et al., 2023b), with the exception that we use multiple templates to address template sensitivity. The prompt templates for law domain are based on Chalkidis (2023). The UNFAIR-ToS (Lippi et al., 2019) task is a multi-label classification task. To get model predictions for this task, we categorize it as a multiple-choice question, and the accuracy of an individual data example is considered true if the model prediction (i.e., the option with the highest per-token likelihood) belongs to the label(s) set. In the biomedicine domain, some classification tasks, including MQP (McCreery et al., 2020), RCT (Dernoncourt & Lee, 2017), and ChemProt (Kringelum et al., 2016), are too challenging for the model, thus we conduct few-shot prompting and maintain the number of demonstrations the same in each class.

**Fine-tuning.** In fine-tuning, we utilize a fixed prompt template (the one displayed in Table 11) for each task to convert data input-output into question-answering pairs. The model is then trained on these pairs for one epoch with the warm-up step as 0. All other training settings are the same with domain-adaptive pre-training. Fine-tuning evaluation is similar to prompting evaluation, but with two differences to align with the fine-tuning training stage: no demonstration is presented before the prompt input, and the prompt template is the same with the one used in the training stage.

Table 10: **Specifications of the domain-specific task datasets.** # Demos is the number of demonstrations in prompting evaluation.

| Task | Type | Metric | # Demos |
|---|---|---|---|
| *BioMed.* | | | |
| MQP (McCreery et al., 2020) | Binary classification | Accuracy | 4 |
| PubMedQA (Jin et al., 2019) | Multi-class classification | Accuracy | 0 |
| USMLE (Jin et al., 2020) | Multi-chioice QA | Accuracy | 0 |
| RCT (Dernoncourt & Lee, 2017) | Multi-class classification | Micro F1 | 10 |
| ChemProt (Kringelum et al., 2016) | Multi-class classification | Micro F1 | 13 |
| *Finance* | | | |
| FiQA SA (Maia et al., 2018) | Multi-class classification | Weighted F1 | 5 |
| FPB (Malo et al., 2014) | Multi-class classification | Weighted F1 | 5 |
| NER (Alvarado et al., 2015) | Named entity recongnition | Entity-level F1 | 20 |
| Headline (Sinha & Khandait, 2020) | Binary-class classification | Weighted F1 | 5 |
| ConvFinQA (Chen et al., 2022) | Question Answering | Exact Match | 0 |
| *Law* | | | |
| SCOTUS (Spaeth et al., 2020) | Multi-class classification | Micro/Macro F1 | 0 |
| CaseHOLD (Zheng et al., 2021) | Multi-chioice QA | Micro/Macro F1 | 0 |
| UNFAIR-ToS (Lippi et al., 2019) | Multi-label classification | Accuracy | 4 |

Table 11: **Prompt templates.** Each template example is paraphrased to multiple variations for prompting evaluation.

| Task | Template |
|---|---|
| *BioMed.* | |
| MQP | Question 1: {QUESTION1}
Question 2: {QUESTION2}
Are questions 1 and 2 asking the same thing? {ANSWER} |
| PubMedQA | Context: {CONTEXT}
Question: {QUESTION}
Answer: {ANSWER} |
| USMLE | Question: {QUESTION}
Answer: {ANSWER} |
| RCT | {SENTENCE}
Question: what is the role of this sentence in an abstract?
Answer: {ANSWER} |
| ChemProt | {SENTENCE}
Question: what is the relation?
Answer: {ANSWER} |
| *Finance* | |
| FiQA SA | {SENTENCE}
Question: what is the sentiment on {TARGET}?
Answer: {ANSWER} |
| FPB | {SENTENCE}
Question: what is the sentiment?
Answer: {ANSWER} |
| NER | {SENTENCE}
Extract named entity: {ANSWER} |
| Headline | {SENTENCE}
Question: {QUESTION}
Answer: {ANSWER} |
| ConvFinQA | {CONTEXT}
{PREVIOUS QAS}
{QUESTION} {ANSWER} |
| *Law* | |
| SCOTUS | Given the following opinion from the Supreme Court of USA (SCOTUS):
"{TEXT}"
The relevant issue area is: {ANSWER} |
| CaseHOLD | Complete the following excerpt from a US court opinion:
{CONTEXT}: {ANSWER} |
| UNFAIR-ToS | Given the following sentence from an online Term of Services:
"{SENTENCE}"
The sentence is unfair with respect to: {ANSWER} |

# E    FURTHER ABLATIONS ON COMPREHENSION TASKS

Figure 3 presents the percentages of mined examples of each task type in all the comprehension task examples, with Word-To-Text, Summarization, and Text Completion accounting for the highest ratios.

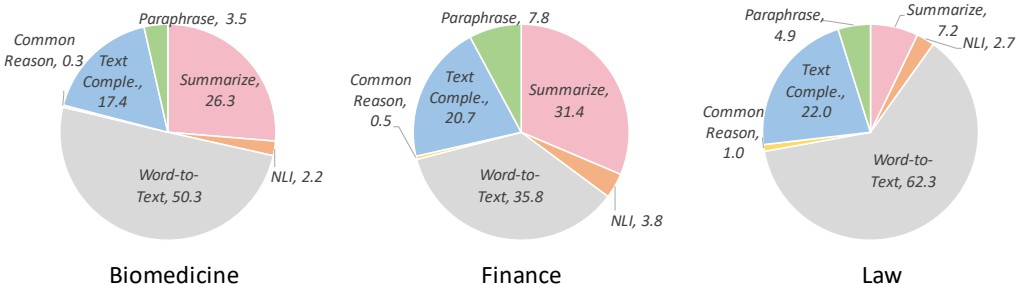

Figure 3: **Percentages of mined examples of each task type in all the comprehension task examples**.

In the biomedicine domain, we conduct ablations on each comprehension task type by systematically removing each task type from the reading comprehension texts. We then use the resulting modified reading comprehension texts to train the general model. Subsequently, we evaluate these trained models on both domain-specific tasks and general LLM benchmarks to analyze the impacts of these ablations.

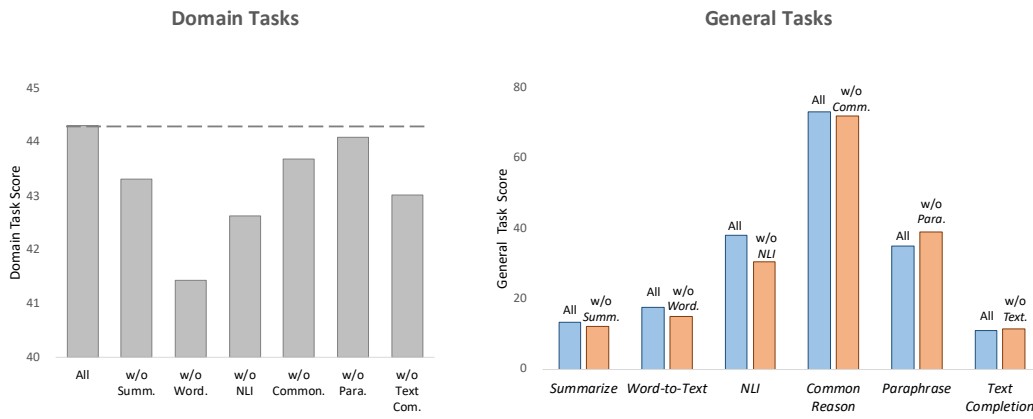

Figure 4: **Prompting scores of domain-specific tasks (left) and general LLM benchmarks (right)** of models trained with different comprehension tasks. **All** denotes the model trained with all the comprehension tasks, while **w/o Summ.** represents the model trained with the comprehension tasks excluding Summarization tasks, **w/o Word.** represents the model trained with the comprehension tasks excluding Word-to-Text tasks, and so on. We report the average task scores within each domain/type.

**Domain-specific Tasks.** As shown in Figure 4, when evaluating on the domain-specific tasks, the removal of any comprehension task type leads to a decrease in task performance, showing their contributions to these domain-specific tasks. Notably, removing Word-to-Text, Summarization, or Text Completion tasks results in a noticeable drop in performance, aligning with the high percentages of these tasks in the mined examples.

Interestingly, even though the Natural Language Inference task type doesn't constitute a large portion of the comprehension tasks, its removal leads to a substantial decrease in performance. This could be attributed to its unique role as the only classification task type within all the comprehension

tasks. In contrast, the impact of removing Commonsense Reasoning and Paraphrase Detection tasks is less pronounced, reflecting their lower percentages in the mined task examples. However, this also suggests the potential benefits of including more diverse task examples, which could further enhance domain-specific task performance.

**General LLM Benchmarks.** Additionally, we conduct experiments where we remove a specific task type from all the comprehension tasks. We then evaluate the trained model's performance specifically on the general tasks corresponding to the removed task type, aiming to demonstrate whether the comprehension tasks have a positive impact on the respective downstream tasks.

In the results for general tasks in Figure 4, when we exclude a particular task type from the comprehension tasks, we observe performance declines in the corresponding downstream tasks, specifically for Summarization, Word-to-Text, Natural Language Inference, and Commonsense Reasoning tasks. This suggests a beneficial connection between the trained comprehension tasks and their corresponding downstream tasks.

However, when we remove Paraphrase Detection or Text Completion, it does not lead to a performance decline in the corresponding tasks. This discrepancy may be attributed to the reformatting of Paraphrase Detection from a classification task to a generation task in the comprehension tasks, causing a mismatch between training and evaluation settings. Furthermore, the Text Completion comprehension task type lacks obvious counterparts in the general LLM benchmarks, which may contribute to the mismatch in the performance change trend.

# F    ANALYSIS OF DOMAIN KNOWLEDGE AND PROMPTING ABILITY

Table 12: **Fine-tuning performance on the domain-specific tasks** of the general large language model (**General LLM**), the model trained on domain-specific raw corpora (**Raw Text**) and the model trained on the reading comprehension texts constructed based on the raw corpora (**Read. Compre.**).

| BioMed. | PubMedQA | ChemProt | MQP | RCT | UMSLE | AVERAGE |
|---|---|---|---|---|---|---|
| General LLM | 75.4 | 64.6 | 55.4 | 87.0 | 38.5 | 64.2 |
| Raw Text | **76.2** | 64.8 | 65.6 | 87.0 | 39.0 | 66.5 |
| Read. Compre. | 76.0 | **65.4** | **87.9** | **87.5** | **41.0** | **71.5** |

| Finance | ConvFinQA | FPB | FiQA SA | Headline | NER | AVERAGE |
|---|---|---|---|---|---|---|
| General LLM | **58.1** | 81.9 | 86.4 | 95.7 | 77.5 | 79.9 |
| Raw Text | 56.2 | 83.3 | **87.9** | 95.8 | 81.3 | 80.9 |
| Read. Compre. | 57.2 | **88.6** | 83.1 | **96.1** | **82.5** | **81.5** |

| Law | SCOTUS | | CaseHOLD | | UNFAIR-ToS | AVERAGE |
|---|---|---|---|---|---|---|
| | mic-F1 | mac-F1 | mic-F1 | mac-F1 | | |
| General LLM | 31.7 | 14.0 | 35.3 | 35.3 | 93.8 | 42.0 |
| Raw Text | 36.7 | **26.0** | 35.4 | 35.4 | 93.7 | 45.4 |
| Read. Compre. | **40.0** | **26.0** | **35.5** | **35.5** | **94.2** | **46.2** |

Table 13: **Prompting results on general LLM benchmarks. Raw** trains on the raw corpora, and **Read** trains on the reading comprehension texts constructed based on the raw corpora. Text Completion is more close to a question type than a task type in general benchmarks, so we report the average of all the tasks following the free-form text completion question type. Each task corresponds to multiple prompt templates taken from FLAN (Wei et al., 2022), and we remove the option suffixes from the templates (Cheng et al., 2023) to fit for the prompting evaluation approach by Brown et al. (2020).

| Task | Metric | General LLM | BioMed. | | Finance | | Law | |
|---|---|---|---|---|---|---|---|---|
| | | | Raw | Read | Raw | Read | Raw | Read |
| *Summarization* | | | | | | | | |
| AGNews (Zhang et al., 2015) | Acc | 58.7 | 51.7 | 55.5 | 56.1 | 50.1 | 57.8 | 60.6 |
| | R-1 | 1.5 | 3.6 | 7.5 | 1.9 | 10.8 | 3.4 | 7.4 |
| AESLC (Zhang & Tetreault, 2019) | R-2 | 0.2 | 0.9 | 2.8 | 0.3 | 3.8 | 0.8 | 2.7 |
| | R-L | 1.5 | 3.6 | 7.2 | 1.8 | 10.3 | 3.3 | 7.2 |
| | R-1 | 0.6 | 3.8 | 9.3 | 3.1 | 13.2 | 3.4 | 9.8 |
| Gigaword (Napoles et al., 2012) | R-2 | 0.1 | 0.7 | 2.4 | 0.6 | 3.8 | 0.6 | 2.5 |
| | R-L | 0.6 | 3.5 | 8.5 | 2.9 | 12.1 | 3.1 | 8.9 |
| *Word-to-Text* | | | | | | | | |
| | R-1 | 14.2 | 16.1 | 24.9 | 17.2 | 24.2 | 17.8 | 27.9 |
| CommonGen (Lin et al., 2020) | R-2 | 0.0 | 1.6 | 5.9 | 1.5 | 6.2 | 2.5 | 7.2 |
| | R-L | 14.2 | 15.4 | 22.2 | 16.2 | 21.1 | 16.7 | 24.3 |
| | R-1 | 17.0 | 21.3 | 20.0 | 21.9 | 22.3 | 23.6 | 19.5 |
| DART (Nan et al., 2021) | R-2 | 3.7 | 6.0 | 5.0 | 6.7 | 7.1 | 7.5 | 5.9 |
| | R-L | 16.0 | 19.6 | 18.5 | 20.1 | 20.5 | 21.4 | 18.0 |
| | R-1 | 14.3 | 18.4 | 27.6 | 22.6 | 37.3 | 18.4 | 28.3 |
| E2ENLG (Dusek et al., 2019) | R-2 | 3.2 | 5.2 | 9.8 | 7.5 | 15.1 | 5.8 | 10.5 |
| | R-L | 13.4 | 17.1 | 24.5 | 20.4 | 32.9 | 17.0 | 25.5 |
| *Natural Language Inference* | | | | | | | | |
| MNLI-m (Williams et al., 2018) | Acc | 35.6 | 35.0 | 36.7 | 36.2 | 37.4 | 34.8 | 34.9 |
| MNLI-mm (Williams et al., 2018) | Acc | 34.9 | 34.9 | 36.1 | 36.8 | 37.2 | 35.7 | 34.4 |
| QNLI (Rajpurkar et al., 2018) | Acc | 51.1 | 52.4 | 53.7 | 52.1 | 52.0 | 50.7 | 51.9 |
| RTE (Bentivogli et al., 2009) | Acc | 30.7 | 9.4 | 30.0 | 23.1 | 26.0 | 18.8 | 32.9 |
| SNLI (Bowman et al., 2015) | Acc | 34.0 | 34.5 | 33.8 | 36.8 | 38.1 | 34.7 | 33.9 |
| *Commonsense Reasoning* | | | | | | | | |
| COPA (Roemmele et al., 2011) | Acc | 71.0 | 68.0 | 75.0 | 74.0 | 73.0 | 72.0 | 74.0 |
| PIQA (Bisk et al., 2020) | Acc | 71.7 | 70.9 | 71.3 | 71.6 | 71.7 | 71.9 | 71.8 |
| HellaSwag (Zellers et al., 2019) | Acc | 71.8 | 71.8 | 72.6 | 72.5 | 73.0 | 72.0 | 72.5 |
| *Paraphrase Detection* | | | | | | | | |
| MRPC (Dolan & Brockett, 2005) | Acc | 35.5 | 31.6 | 34.8 | 33.3 | 37.3 | 34.1 | 36.0 |
| | F1 | 17.0 | 0.0 | 13.6 | 9.9 | 22.4 | 8.8 | 19.7 |
| QQP (Wang et al., 2019) | Acc | 56.0 | 62.7 | 61.1 | 52.5 | 54.3 | 60.6 | 59.9 |
| | F1 | 34.0 | 3.0 | 10.7 | 33.1 | 44.5 | 15.3 | 14.4 |
| Paws Wiki (Zhang et al., 2019) | Acc | 54.8 | 55.6 | 54.9 | 53.7 | 53.7 | 55.4 | 54.8 |
| *Closed-book QA* | | | | | | | | |
| ARC-c (Bhakthavatsalam et al., 2021) | Acc | 37.3 | 36.7 | 39.0 | 37.8 | 38.8 | 37.6 | 39.3 |
| ARC-e (Bhakthavatsalam et al., 2021) | Acc | 58.5 | 59.2 | 63.1 | 59.4 | 62.3 | 60.0 | 62.9 |
| NQ (Kwiatkowski et al., 2019) | EM | 2.2 | 0.1 | 0.5 | 0.1 | 0.0 | 0.1 | 0.1 |
| | F1 | 3.4 | 0.8 | 1.9 | 1.2 | 2.1 | 1.3 | 1.8 |
| CMSQA (Talmor et al., 2019) | Acc | 39.6 | 40.4 | 43.5 | 40.0 | 42.6 | 40.9 | 44.6 |
| *Reading Comprehension* | | | | | | | | |
| BoolQ (Clark et al., 2019) | Acc | 55.7 | 42.6 | 50.7 | 53.4 | 55.8 | 51.0 | 53.9 |
| OBQA (Mihaylov et al., 2018) | Acc | 46.2 | 45.8 | 46.4 | 46.0 | 47.0 | 45.4 | 47.2 |
| SQuADv1 (Rajpurkar et al., 2016) | EM | 0.1 | 1.0 | 3.5 | 0.2 | 0.0 | 0.3 | 2.0 |
| | F1 | 0.2 | 4.6 | 10.0 | 0.6 | 7.3 | 6.3 | 10.7 |
| MultiRC (Khashabi et al., 2018) | Acc | 55.9 | 47.5 | 53.0 | 49.9 | 49.7 | 52.5 | 52.4 |

## G CASES OF READING COMPREHENSION TEXTS

**Raw Text**

**Here is the first part of an article about biomedicine:** Recent reported evidence indicates that vocal cord carcinoma is evolving similarly to oropharyngeal cancer with an increasing number of patients without a smoking history having human papillomavirus (HPV) disease. Observations also suggest that an increasing number of patients who present with glottic carcinoma are younger than has been reported in the past. **(…TL; DR...)**

**Summarization**

However, recent published evidence shows that glottic carcinoma can be an HPV-related disease with increasing incidence in nonsmokers. It isn't surprising that alternate malignant pathways may have a different timeline.

Answer questions based on the article:

**What is a summary?** Glottic Carcinoma in Young Patients.

**Word-to-Text**

**Generate a sentence that includes these biomedicine words [carcinoma, oropharyngeal, papillomavirus]:** Recent reported evidence indicates that vocal cord carcinoma is evolving similarly to oropharyngeal cancer with an increasing number of patients without a smoking history having human papillomavirus (HPV) disease.

**Natural Language Inference**

**Premise: Observations also suggest that an increasing number of patients who present with glottic carcinoma are younger than has been reported in the past.**

**Hypothesis: An investigation was done to examine the incidence of glottic carcinoma in patients 30 years old (y/o) or younger.**

**Does the premise entail the hypothesis?** Yes

**Commonsense Reasoning**

**What is the reason for "None of the 11 glottic cancer patients had been previously treated for benign recurrent respiratory papillomatosis although it was initially suspected prior to biopsy"?** The morphology of the lesions and the patients' young age.

**Paraphrase Detection**

**Compose a sentence that contradicts the meaning of "Historically, glottic carcinoma is considered to be a tobacco-induced disease.".**

**Answer:** Recent published evidence shows that glottic carcinoma can be an HPV-related disease with increasing incidence in nonsmokers.

**Text Completion**

**How would you complete the article?** This finding further supports the concept that glottic carcinoma is an evolving disease, and it demonstrates the increasing importance of discriminating potential glottic carcinomas in young patients from benign low-risk HPV recurrent respiratory papillomatosis.

Figure 5: **An example of a reading comprehension text constructed from a raw text.** The underlined sentence is added to guide the model to answer questions based the given context.

Table 14: **A case of a reading comprehension text in biomedicine domain.** Certain portions are omitted for brevity and are represented as **(...)**.

---

Pancreastatin (PST), a chromogranin A-derived peptide, has been found to modulate glucose, lipid, and protein metabolism in rat adipocytes. PST has an overall counterregulatory effect on insulin action by activating a specific receptor-effector system (Galpha(q/11) protein-PLC-beta-PKC(classical)). However, PST stimulates both basal and insulin-mediated protein synthesis in rat adipocytes. In order to further investigate the mechanisms underlying the effect of PST stimulating protein synthesis, we sought to study the regulation of different components of the core translational machinery by the signaling triggered by PST. Thus, we studied ribosomal p70 S6 kinase, phosphorylation of the cap-binding protein (initiation factor) eIF4E, and phosphorylation of the eIF4E-binding protein 4E-BP1 (PHAS-I). We have found that PST stimulates the S6 kinase activity, as assessed by kinase assay using specific immunoprecipitates and substrate. This effect was checked by Western blot with specific antibodies against the phosphorylated S6 kinase. Thus, PST dose-dependently stimulates Thr421/Ser424 phosphorylation of S6 kinase. Moreover, PST promotes phosphorylation of regulatory sites in 4E-BP1 (PHAS-I) (Thr37, Thr46). The initiation factor eIF4E itself, whose activity is also increased upon phosphorylation, is phosphorylated in Ser209 by PST stimulation. **(...)**
Use evidence from the biomedicine article to answer these questions:

Assess the relationship between Sentence 1: "This effect was checked by Western blot with specific antibodies against the phosphorylated S6 kinase."
Sentence 2: "PST dose-dependently stimulates Thr421/Ser424 phosphorylation of S6 kinase."
Is it characterized as Entailment, Neutral, or Contradiction? Entailment

Assess the relationship between Sentence 1: "PST has an overall counterregulatory effect on insulin action by activating a specific receptor-effector system (Galpha(q/11) protein-PLC-beta-PKC(classical))."
Sentence 2: "PST stimulates both basal and insulin-mediated protein synthesis in rat adipocytes."
Is it characterized as Entailment, Neutral, or Contradiction? Contradiction

Next question: What is the reason of the following sentence?
We studied ribosomal p70 S6 kinase, phosphorylation of the cap-binding protein (initiation factor) eIF4E, and phosphorylation of the eIF4E-binding protein 4E-BP1 (PHAS-I). In order to further investigate the mechanisms underlying the effect of PST stimulating protein synthesis, we sought to study the regulation of different components of the core translational machinery by the signaling triggered by PST.

Next question: What is the reason of the following sentence?
PST dose-dependently stimulates Thr421/Ser424 phosphorylation of S6 kinase. This effect was checked by Western blot with specific antibodies against the phosphorylated S6 kinase.

"PST has an overall counterregulatory effect on insulin action by activating a specific receptor-effector system (Galpha(q/11) protein-PLC-beta-PKC(classical))." Generate a sentence that expresses a contrasting idea to the previous statement. PST stimulates both basal and insulin-mediated protein synthesis in rat adipocytes.

Here are some concepts about biomedicine: mechanisms, regulation, translational.
Generate a detailed description of them In order to further investigate the mechanisms underlying the effect of PST stimulating protein synthesis, we sought to study the regulation of different components of the core translational machinery by the signaling triggered by PST.

Here are some concepts about biomedicine: phosphorylation, stimulation, phosphorylated.
Generate a detailed description of them The initiation factor eIF4E itself, whose activity is also increased upon phosphorylation, is phosphorylated in Ser209 by PST stimulation.

Briefly summarize this text. Pancreastatin, a chromogranin A-derived peptide, activates protein synthesis signaling cascade in rat adipocytes.

---

Table 15: **A case of a reading comprehension text in finance domain.** Certain portions are omitted for brevity and are represented as **(...)**.

---

Read the beginning of an article on finance: In this article, we discuss the 12 biggest commercial janitorial companies in USA. If you want to skip our detailed analysis of these companies, go directly to the 5 Biggest Commercial Janitorial Companies In USA. According to Statista, the janitorial services sector's market size will increase by 6.6 percent in 2022. The annualized percentage of this market's growth was 6.7% between 2017 and 2022. Additionally, between 2020 and 2021, the number of janitors and cleaners employed in the United States rose by nearly 50,000. By 2024, it is predicted that the U.S. janitorial services industry will generate around $50.1 billion in sales. The market size, measured by revenue, of the janitorial services industry is $97.6 billion in 2022. Businesses frequently grow when corporate profits increase, raising demand for janitorial services. However, dwindling corporate profit decreases demand for janitorial services as companies close facilities and cut back on the frequency of contracted cleaning to cut expenses. The industry may have an opportunity in 2022 as corporate profits are anticipated to rise, according to ibisworld. According to Allied Market Research, the cleaning services market is to reach $111.49 billion globally by 2030 at a 6.5% CAGR, as reported by Bloomberg. The global cleaning services industry is expanding due to service providers expanding their online presence and rising commercial consumer demand. However, heightened rivalry and the introduction of new companies limit market expansion. On the other hand, the demand for construction and post-construction cleaning services will open up new prospects. The Covid-19 pandemic has had a favorable effect on the global cleaning services business. Deep cleaning and disinfection have become more popular among residential and commercial users to reduce the risk of cross-contamination. **(...)**

How would you extend the article? Moore adds that Casella Waste Systems, Inc. (NASDAQ:CWST) is also the only public waste company its size not to be acquired, which provides "downside support to the valuation on a takeout potential". 09. Stericycle, Inc. (NASDAQ: SRCL) Market Capitalization as of November 18, 2022: $4.73 billion Stericycle, Inc. (NASDAQ:SRCL) was incorporated in 1989 and is located in Bannockburn, Illinois. **(...)**

Then, answer the following questions based on the whole article:

Please write a short summary for the above article. 12 Biggest Commercial Janitorial Companies In USA

Given the sentence: The global cleaning services industry is expanding due to service providers expanding their online presence and rising commercial consumer demand.
Select the most suitable word to connect it with: "Heightened rivalry and the introduction of new companies limit market expansion."
Options: Therefore, However, Maybe However

Answer the following question. The global cleaning services industry is expanding due to? Service providers expanding their online presence and rising commercial consumer demand.

Given the sentence "Businesses frequently grow when corporate profits increase, raising demand for janitorial services.", come up with a different sentence that contradicts its meaning. Dwindling corporate profit decreases demand for janitorial services as companies close facilities and cut back on the frequency of contracted cleaning to cut expenses.

Given the sentence "The global cleaning services industry is expanding due to service providers expanding their online presence and rising commercial consumer demand.", come up with a different sentence that contradicts its meaning. Heightened rivalry and the introduction of new companies limit market expansion.

Generate a sentence that includes the following finance keywords: Capitalization, Incorporated, Industries ABM Industries Incorporated (NYSE: ABM) Market Capitalization as of November 18, 2022: $2.98 billion ABM Industries Incorporated (NYSE:ABM) was incorporated in 1985 and is based in New York, New York.

---

Table 16: **A case of a reading comprehension text in law domain.** Certain portions are omitted for brevity and are represented as **(...)**.

---

Here is the first part of an article about law: The district court ordered Arledge to pay restitution in the amount of \$5,829,334.90, without interest, to the Settlement Fund pursuant to the Mandatory Victims Restitution Act of 1996 ("MVRA"), 18 U.S.C. §3663A. Arledge disputes the calculation used to determine the amount of loss, claiming that the government did not sufficiently prove that the Settlement Fund had paid false claims arising from Arledge's illegal conduct. Specifically, he objects to the calculation of losses related to the Fen Phen II settlement.

The "general rule is that a district court can award restitution to victims of the offense, but the restitution award can encompass only those losses that resulted directly from the offense for which the defendant was convicted." United States v. Maturin, 488 F.3d 657, 660-61 (5th Cir. 2007) (citing Hughey v. United States, 495 U.S. 411, 413, 110 S.Ct. 1979, 109 L.Ed.2d 408 (1990)). The pre-sentence report attributed forty-seven fraudulent claims to the offenses for which Arledge was convicted. There were three categories of evidence used to substantiate the government's assertion that these claims resulted from Arledge's illegal conduct: (1) the testimony of Wyatt, an employee of S&A, who created fraudulent documents; **(...)**

Please complete it: We have stated repeatedly that an order of restitution must be limited to losses caused by the specific conduct underlying the offense of conviction. See United States v. Griffin, 324 F.3d 330, 367 (5th Cir.2003) (holding that restitution is restricted to the limits of the offense); Tencer, 107 F.3d at 1135-36. Accordingly, we decline the government's request to adopt a harmless error analysis for the calculation of loss under the MVRA, and we remand for a re calculation of actual loss based upon the evidence in the record.

Then, answer the following questions based on the whole article:

Based on the statement Arledge makes no convincing argument regarding why the identification by AHP is not sufficient or how the district court abused its discretion in relying upon the representations by AHP., provide a continuation using the word "Hence" to express the following idea. It was not an abuse of discretion for the court to award restitution that encompassed those losses resulting from the creation of fraudulent documents in furtherance of the scheme to defraud for which Arledge was convicted.
**(...)**
What happens before "The government asks this court to find that the failure to provide proof of these three claims was harmless error."? It also asserts that it would be able to provide, on remand, proof of an additional false claim that was not originally included in the district court's restitution order, which resulted in a loss of \$488,000.

In opposition to the sentence "Thus, it was not an abuse of discretion for the court to award restitution that encompassed those losses resulting from the creation of fraudulent documents in furtherance of the scheme to defraud for which Arledge was convicted.", create a sentence with a contradictory meaning. There are three claims for which the government concedes that there was no proof of falsity: Florestine Baker (\$18,000), Shirley E.

Produce a long descriptive sentence about law that uses all these words: prescription, manufactured, fraudulent. There were three categories of evidence used to substantiate the government's assertion that these claims resulted from Arledge's illegal conduct: (1) the testimony of Wyatt, an employee of S&A, who created fraudulent documents; (2) the testimony of two pharmacists who testified that specific prescriptions allegedly from their pharmacies were, in fact, manufactured; and (3) representations by AHP that the claims were fraudulent.

Produce a long descriptive sentence about law that uses all these words: restitution, fraudulent, furtherance. Thus, it was not an abuse of discretion for the court to award restitution that encompassed those losses resulting from the creation of fraudulent documents in furtherance of the scheme to defraud for which Arledge was convicted.

What was this article about? Amount of Loss

---

# H  APPLICABILITY TO OTHER MODELS

To assess the effectiveness of our method across different language models, such as smaller or larger models, models of various types, and those that have undergone reinforcement learning from human feedback (RLHF) (Christiano et al., 2017), we apply our method to the following models: Pythia-70M (Biderman et al., 2023), LLaMA-13B (Touvron et al., 2023a), and LLaMA-2-Chat-7B (Touvron et al., 2023b). The experimental results presented in Table 17 demonstrate consistent improvements achieved by our methods across various models in the biomedicine domain.

Table 17: **Domain-specific task performance of different language models.**

|  | Pythia-70M | | LLaMA-13B | | LLaMA-2-Chat-7B | |
|---|---|---|---|---|---|---|
|  | General LLM | AdaptLLM | General LLM | AdaptLLM | General LLM | AdaptLLM |
| PubMedQA | 34.4 | **50.3** | 59.6 | **66.0** | 55.2 | **65.0** |
| ChemProt | 7.6 | **10.4** | 42.8 | **47.6** | 33.8 | **36.6** |
| MQP | **52.1** | 50.5 | 49.3 | **73.0** | 59.3 | **60.3** |
| RCT | 29.6 | **30.6** | **56.7** | 50.4 | 53.4 | **59.2** |
| USMLE | 24.0 | **24.4** | **34.7** | 34.0 | 29.1 | **33.4** |
| AVERAGE | 29.5 | **33.2** | 48.6 | **54.2** | 46.2 | **50.9** |

**Smaller and Different: Pythia-70M.** Pythia-70M (Biderman et al., 2023) is a recently released and representative language model with 70 million parameters. It shares the same architecture as GPT-NeoX (Black et al., 2022) and can also serve as a model with a different architecture than LLaMA.

**Larger: LLaMA-13B.** We scale up our base model to the one with 13 billion parameters. Due to computational constraints, we reduce the maximum sequence length to 512 tokens (1/4 of AdaptLLM-7B) during continued training. To maintain the same number of trained tokens, we increase the continued training steps to 40k (4 times that of AdaptLLM-7B).

**After RLHF: LLaMA-2-Chat.** We apply our method to LLaMA-2-Chat-7B (Touvron et al., 2023b), which has undergone supervised fine-tuning and reinforcement learning with human feedback (Christiano et al., 2017) to align with human preferences. LLaMA-2-Chat requires specific data formats; an example of the data format provided by Hugging Face [4] is shown in Figure 6.

```
[INST] <<SYS>>
{SYSTEM PROMPT}
<</SYS>>

{USER_MSG_1} [/INST] {MODEL_ANSWER_1} [INST] {USER_MSG_2}} [/INST] {MODEL_ANSWER_1}
```

Figure 6: **Input data format for LLaMA-2-Chat.**

We observe that our reading comprehension can seamlessly conform to the required data format by transforming the reading comprehension into a multi-turn conversation. For example, we can convert the reading comprehension shown in Figure 1 into the conversation format depicted in Figure 7, where we employ the system prompt from the Hugging Face LLaMA-2 chat demo.

Note that during the training of LLaMA-2-Chat, each input in the training data comprises only one piece of single/multi-turn conversation, because we empirically find that concatenating multiple conversations as one data input would lead to uncontrollable gradient overflow, which would quickly shut down the training. In the evaluation stage, we adhere to this data format and transform each few-shot prompt into a multi-turn conversation.

---

[4] https://huggingface.co/blog/llama2

[INST] <<SYS>>

You are a helpful, respectful and honest assistant. Always answer as helpfully as possible, while being safe. Your answers should not include any harmful, unethical, racist, sexist, toxic, dangerous, or illegal content. Please ensure that your responses are socially unbiased and positive in nature.

If a question does not make any sense, or is not factually coherent, explain why instead of answering something not correct. If you don't know the answer to a question, please don't share false information.

<</SYS>>

Here is the first part of an article about biomedicine: Recent reported evidence indicates that vocal cord carcinoma is evolving similarly to oropharyngeal cancer with an increasing number of patients (...)

Answer questions based on the article:

What is a summary? [/INST] Glottic Carcinoma in Young Patients.

[INST]

Generate a sentence that includes these biomedicine keywords [carcinoma, oropharyngeal, papillomavirus]: [/INST] Recent reported evidence indicates that vocal cord carcinoma is evolving...

[INST]

Premise:... Hypothesis:... Does the premise entail the hypothesis? [/INST] Yes

[INST]

What is the reason for "..."? [/INST] the morphology of the lesions and the patients' young age.

[INST]

Compose a sentence that contradicts the meaning of "Historically, glottic carcinoma ... ".

Answer: [/INST] Recent published evidence ...

[INST]

How would you complete the article? [/INST] This finding further supports...

Figure 7: **An example of transforming reading comprehension into a multi-turn conversation for LLaMA-2-Chat.**

## I   ANALYSIS ON VERBALIZER EFFECT

We conduct an analysis on how the diversity of verbalizers influences performance. Intuitively, our hypothesis is that a greater diversity of verbalizers leads to a broader range of comprehension tasks, ultimately enhancing performance. To test this hypothesis, we conduct an ablation by preserving only one verbalizer for each task type. Specifically, we preserve only the first verbalizer in each row in Table 3, and remake the reading comprehension data to observe the effect.

First, the most direct impact is that the number of mined task examples decreases. When using all the verbalizers, we mine an average of 2.1 examples per text, while using one verbalizer results in 1.4 examples per text. Next, we conduct an experiment to train the general model using the new data, and the task results shows that with fewer verbalizers, the downstream task performance declines (from 44.1 to 42.7), verifying that including more verbalizers can contribute to better performance.

## J   ANALYSIS ON DOMAIN-SPECIFIC RAW CORPORA

As shown in the experiment results in Section 2, continued training on domain-specific raw corpora leads to decreased prompting ability. To verify whether this reduction results from the limited diversity of input-output patterns within the domain-specific corpora, we conduct a case study on the question answering pattern. Specifically, we search over all three domain-specific corpora and a representative corpora in the general domain—RedPajama (Computer, 2023)—to estimate how many sentence pairs follow the question-answering pattern, utilizing this mining pattern:

```
{What|Who|When|Where|Why|Which|How}{SENT1}? {SENT2}.
```

Table 18 lists the number of tokens belonging to the mined question-answering pairs per million tokens. Compared with the general corpora, domain-specific corpora have much lower rates of data following the question-answering pattern, which could lead to the observed decrease in question-answering ability.

Table 18: **Number of tokens belonging to the mined question-answering pairs** per million tokens of different pre-training corpora.

| Pre-train Corpora | BioMed. | Law | Finance | General |
|---|---|---|---|---|
| # QA tokens | 22.5 | 0.8 | 236.8 | **490.26** |

