# OpenReview forum: "Adapting Large Language Models via Reading Comprehension"
_ICLR.cc/2024/Conference — ICLR 2024 poster_

### Official Review · Reviewer_pD3t · 2023-10-31

**Soundness:** 3 good
**Presentation:** 3 good
**Contribution:** 2 fair
**Rating:** 6
**Confidence:** 4

**Summary:**

The paper reveals that domain-specific pre-training greatly reduces LLMs'  prompting ability. The authors introduce a method to transform raw texts into comprehension tasks, aiming to enhance LLM's domain knowledge without sacrificing prompting skills. Experiment results show their 7B model is competitive with larger models.

**Strengths:**

1. Interesting observation: Pre-training on domain-specific datasets leads to a decline in LLM's prompting ability.
2. The proposed idea is straightforward, simple, and has the potential for broad applicability with clear motivation.
3. Experimental findings indicate that domain-specific reading comprehension texts enhance the model's performance.

**Weaknesses:**

1. The novelty seems somewhat constrained, especially regarding the idea of incorporating reading comprehension tasks during the pre-training phase, which appears similar to the following paper:
RECKONING: Reasoning through Dynamic Knowledge Encoding.
2. The authors conducted experiments only on the 7B model. It's uncertain whether consistent results would be observed on larger-scale models, and it's unclear if the proposed method is effective on models after RLHF. (However, I think this weakness doesn't undermine the contribution.)

**Questions:**

I'm confused why pre-training on domain-specific knowledge leads to a decrease in LLM's prompting ability. Could you clarify?
Especially in the explanation, does the term 'input-output patterns' pertain to limited data patterns or is it referring to something else?

---

> ### Author Response · Authors · 2023-11-21
> **Response to Reviewer pD3t**
>
> Thank you for recognizing the interesting observations in our paper and acknowledging the effectiveness of our method. In response to your questions regarding the comparison with RECKONING [1], other models, and the decrease in prompting ability, we provide the following analysis.
>
> ### Q1: Comparison with RECKONING [1]
> Thank you for providing such a valuable reference. Below is a comparison between our work with RECKONING, and we have incorporated RECKONING into our paper.
>
> RECKONING enhances reasoning ability by incorporating contextual knowledge into the model's parameters before exposing it to a question at inference time. If we view the model's learning from contextual knowledge as the "reading" process and the model's predicting on the question as the "comprehension" process, the overall RECKONING process can be viewed as a form of reading comprehension.
>
> However, our reading comprehension differs from RECKONING in many aspects:
>
> - The most obvious difference lies in the goal: RECKONING enhances reasoning ability, while we improve domain expertise.
> - Another difference is the timing of model parameter updates: RECKONING performs these updates at inference time, whereas we perform updates during the pre-training stage.
> - Moreover, if I understand it correctly, RECKONING learns contextual knowledge specifically for ad-hoc single or multiple related questions (the number of knowledge-related questions is set up to 18 in the experiment), while we learn domain-specific knowledge for various downstream tasks.
>
> ### Q2: Uncertain whether consistent results would be observed on larger-scale models and models after RLHF.
> Many thanks to your inspirational suggestion! Please see the General Response where **we conduct experiments on a larger model---LLaMA-13B**. Moreover, we find our reading comprehension is well-suited to the data format of **the model after RLHF, contributing to significant gains!**
>
> ### Q3: Why pre-training on domain-specific knowledge leads to a decrease in LLM's prompting ability. Does the term 'input-output patterns' pertain to limited data patterns?
>
> Thanks again for your insightful advice! Your explanation of the term "input-output patterns" is right, it relates to limited data patterns in the domain-specific raw corpora.
>
> However, we'd like to clarify that it is not the domain-specific knowledge that hurts the prompting ability for question answering; instead, **it is the limitation of data patterns within the domain-specific raw corpora that impacts this ability**. We provide an analysis on the data patterns of domain-specific corpora below, and this analysis has also been included in our paper.
>
> **Domain-Specific Corpora Have Fewer QAs than General Corpora.**
> We conduct a case study on the question answering pattern. Specifically, we search over all three domain-specific corpora and a representative corpora in the general domain---RedPajama [2]---to estimate how many sentence pairs follow the question-answering pattern, utilizing this mining pattern:
>
> ```text
> {What|Who|When|Where|Why|Which|How}{SENT1}? {SENT2}
> ```
> The following table lists the number of tokens belonging to the mined question-answering pairs per million tokens. Compared with the general corpora, domain-specific corpora have much lower rates of data following the question-answering pattern, which could lead to the observed decrease in question-answering ability.
>
> |Pre-train Corpora | BioMed | Law | Finance | General (RedPajama) |
> |---------------------------------|--------|-----|---------|----------------------|
> | # QA tokens  | 22.5   | 0.8 | 236.8   | **490.26**           |
>
>
> ### References:
>
> [1] Zeming Chen et al. RECKONING: reasoning through dynamic knowledge encoding. CoRR, abs/2305.06349, 2023.
>
> [2] Together Computer. Redpajama: an open dataset for training large language models, 2023.

---

### Official Review · Reviewer_qcCv · 2023-10-31

**Soundness:** 3 good
**Presentation:** 3 good
**Contribution:** 3 good
**Rating:** 8
**Confidence:** 3

**Summary:**

This paper proposes to adapting LLMs via reading comprehension for QA. The authors conduct extensive experiments on various QA datasets. The results show the effectiveness of the proposed method. The paper is well written and the solution is clear.

**Strengths:**

1. The authors conduct extensive experiments on various QA datasets. The results show the effectiveness of the proposed method.
2.  The paper is well written and the solution is clear.

**Weaknesses:**

1. I download the Supplementary Material and find that there are many missing files, such as codes and the full data sets.
2. The implemtation details are not clear, such as GPU and memory size and  the parameters.

**Questions:**

1. I download the Supplementary Material and find that there are many missing files, such as codes and the full data sets.
2. The implemtation details are not clear, such as GPU and memory size and  the parameters.

---

> ### Author Response · Authors · 2023-11-21
> **Response to Reviewer qcCv**
>
> Thank you very much for your great appreciation of our work! Below, we provide clarifications on the supplementary materials and implementation details.
>
> ### Q1: More Supplementary Materials
> Thank you for the feedback on our supplementary material. In the original supplementary material, we included the code for transferring raw texts to reading comprehension texts, and the evaluation datasets. To address your concerns, **we provide more code and data through this [anonymous link](https://anonymous.4open.science/r/anonymous_supp).**
>
> ### Q2: Implementation Details
> We appreciate your feedback. The implementation details, including the use of 32 V100-32GB GPUs, were presented in Appendix B. To enhance accessibility for readers, **we have incorporated more explicit descriptions in the main body of our paper**, making it easier for the readers to locate the implementation details.

---

### Official Review · Reviewer_aRRU · 2023-11-01

**Soundness:** 4 excellent
**Presentation:** 3 good
**Contribution:** 4 excellent
**Rating:** 6
**Confidence:** 4

**Summary:**

The paper focuses on the investigating domain-adaptation pretraining method for LLMs, where they continued training on domain-specific corpora and found the approach hurts the LLMs' prompting ability. Thus, the author proposes to convert the corpora into reading comprehension texts to preserve the prompting performance. The experiments show the effectiveness of the method in three different domains (biomedicine, law, and finance).

**Strengths:**

- The proposed prompts are very effective, as showed with a significant zero-shot performance improvement across 3 domains, and the 7B model used in the experiment can outperform larger models (50B).
- The paper is well-written. The author puts a comprehensive details on the experiments and analysis.

**Weaknesses:**

- Some ablation studies are essential to understand the effectiveness of the components (e.g., how the verbalizer affects the performance)
- The baselines are not comparable to the results reported by the authors. E.g., GPT-J 6B. It will be great to have the AdaptLLM result on top of the baselines wherever the models are publicly available.

**Questions:**

- How did you choose the set of strings for the verbalizer?
- Are there any analyses or findings on why the model performance improvement on domain "law" is the least?
- Does the approach benefit another type of models? e.g., encoder-decoder model? or other decoder-only models?

---

> ### Author Response · Authors · 2023-11-21
> **Response to Reviewer aRRU (Part 1)**
>
> Many thanks for your appreciation of our soundness and contribution! We hope the following analysis can address your questions about the verbalizer, baseline, law domain improvement, and another type of model.
>
> ### Q1: How did you choose the verbalizer? How the verbalizer affects the performance?
> Thanks for your in-depth suggestion, we conduct the following analysis.
>
> **Verbalizer Selection:**
> Our choice of verbalizers for the Natural Language Inference (NLI) task type aligns with [1], where the effectiveness has been verified. Then, we observe that the entailment relationship in NLI can be extended to other task types like Commonsense Reasoning and Paraphrase Detection. In Table 3 in our paper, you may observe that the verbalizers for NLI are also applied to Commonsense Reasoning and Paraphrase Detection. Finally, we complement verbalizers for the remaining task types based on commonsense; for example, a verbalizer like "is defined as" can be commonly accepted to connect a concept with its definition.
>
>
> **Verbalizer Effect:**
> Intuitively, our hypothesis is that **a greater diversity of verbalizers leads to a broader range of comprehension tasks, ultimately enhancing performance**. To test this hypothesis, we conduct an ablation by preserving only one verbalizer for each task type. Specifically, we preserve only the first verbalizer in each row in Table 3, and remake the reading comprehension data to observe the effect.
>
> First, **the most direct impact is that the number of mined task examples decreases**. When using all the verbalizers (All Verbal), we mine an average of 2.1 examples per text, while using one verbalizer (Single Verbal) results in 1.4 examples per text.
>
>
> |                                | All Verbal | Single Verbal |
> |--------------------------------|--------------------------|-----------------------------|
> | # Mined examples per text | **2.1**                     | 1.4                         |
>
> Next, we conduct an experiment to train the general model using the new data, and the task results are as follows:
>
> |          | All Verbal | Single Verbal |
> |----------|:------------------------:|:---------------------------:|
> | Average Score  | **44.3**                 | 42.7                        |
>
> These results demonstrate that with fewer verbalizers, the downstream task performance declines, verifying that **including more verbalizers can contribute to better performance**.
>
> ### Q2: The baselines are not comparable to the results reported by the authors. E.g., GPT-J 6B. It will be great to have the AdaptLLM result on top of the baselines wherever the models are publicly available.
>
> We included GPT-J-6B as one baseline to facilitate a comparison with LexGPT-6B, which was trained from GPT-J-6B, and LexGPT-6B was the only publicly available large language model in the English-language law domain when we submitted our paper. To address your concern, we conduct experiments on GPT-J-6B and our method shows a positive average score compared to GPT-J-6B:
>
> |                 | GPT-J-6B | LexGPT-6B | AdaptLLM-6B |
> |-----------------|:--------:|:---------:|:-----------:|
> | SCOTUS-mic-F1   | 15.9     | 16.9      | 18.8        |
> | SCOTUS-mac-F1   | 13.6     | 7.7       | 20.1        |
> | CaseHOLD-mic-F1 | 34.9     | 27.0      | 34.7        |
> | CaseHOLD-mac-F1 | 34.9     | 27.0      | 34.7        |
> | UNFAIR-ToS      | 79.8     | 81.9      | 80.0        |
> | AVERAGE         | 35.9     | 32.1      | **37.7**    |
>
> Considering that you may also be curious about our comparison with another baseline, MedAlpaca-13B, which is trained from LLaMA-13B, we conduct experiments on LLaMA-13B and observe favorable results.
>
> |          | LLaMA-13B | MedAlpaca-13B | AdaptLLM-13B |
> |----------|:---------:|:-------------:|:------------:|
> | PubMedQA |   59.6    |     60.7      |   62.5   |
> | ChemProt |   42.8    |     38.4      |   44.4   |
> | MQP      |   49.3    |     57.4      |   72.8   |
> | RCT      |  56.7 |     51.3      |     54.4     |
> | USMLE    |   34.7    |    39.0   |     34.6     |
> | AVERAGE  |   48.6    |     49.4      |   **53.7**   |
>
>
> We've updated Table 4 in our paper, so now that we can achieve the great goal to **have the AdaptLLM result on top of the baselines wherever the models are publicly available**!
>
> (continue in part 2)

---

> ### Author Response · Authors · 2023-11-21
> **Response to Reviewer aRRU (Part 2)**
>
> ### Q3: Why the performance improvement on domain "law" is the least?
>
> Thanks for your careful review, but actually we do not have a very definitive explanation for this phenomenon. By comparing pre-training data in the three domains, we infer that the **low readability of law pre-training corpora** might be the reason:
>
> - Biomedicine---PubMed Abstract [2]: Each text is an abstract for a research paper, which means the text should be highly clear, concise and readable. A randomly sampled example is as follows:
>
> ```text
> Pattern-recognition receptors: signaling pathways and dysregulation in canine chronic enteropathies-brief review.
> Pattern-recognition receptors (PRRs) are expressed by innate immune cells and recognize pathogen-associated molecular patterns (PAMPs) as well as endogenous damage-associated molecular pattern (DAMP) molecules. With a large potential for synergism or convergence between their signaling pathways, PRRs orchestrate a complex interplay of cellular mediators and transcription factors, and thus play a central role in homeostasis and host defense. Aberrant activation of PRR signaling, mutations of the receptors and/or their downstream signaling molecules, and/or DAMP/PAMP complex-mediated receptor signaling can potentially lead to chronic auto-inflammatory diseases or development of cancer. PRR signaling pathways appear to also present an interesting new avenue for the modulation of inflammatory responses and to serve as potential novel therapeutic targets. Evidence for a dysregulation of the PRR toll-like receptor (TLR)2, TLR4, TLR5, and TLR9, nucleotide-binding oligomerization domain-containing protein (NOD)2, and the receptor of advanced glycation end products (RAGE) exists in dogs with chronic enteropathies. We describe the TLR, NOD2, and RAGE signaling pathways and evaluate the current veterinary literature-in comparison to human medicine-to determine the role of TLRs, NOD2, and RAGE in canine chronic enteropathies.
> ```
>
> - Finance---Stock News: We collect stock news using the codebase of FinGPT [3], which could ensure the data quality. A randomly sampled example is as follows:
>
> ```text
> Chevron, Johnson & Johnson share gains lead Dow's 100-point jump
> Powered by strong returns for shares of Chevron and Johnson & Johnson, the Dow Jones Industrial Average is up Wednesday morning. Shares of Chevron CVX, -2.35% and Johnson & Johnson JNJ, -0.13% are contributing to the blue-chip gauge's intraday rally, as the Dow DJIA, -1.01% is trading 105 points higher (0.3%). Chevron's shares have climbed $3.84, or 2.4%, while those of Johnson & Johnson are up $2.85, or 1.8%, combining for an approximately 44-point boost for the Dow. Merck MRK, -0.25%, Apple Inc. AAPL,, and Intel INTC, -1.95% are also contributing significantly to the gain. A $1 move in any one of the 30 components of the benchmark equates to a 6.59-point swing. Editor's Note: This story was auto-generated by Automated Insights, an automation technology provider, using data from Dow Jones and FactSet. See our market data terms of use.
> ```
>
> - Law—--FreeLaw Opinions [2]: Each text is a legal opinion from federal and state courts, documenting many oral snippets. In contrast, the corpora in the other domains mainly consist of written passages. We infer that the unique feature of incorporating spoken language may affect the overall readability of law corpora. A randomly sampled example is as follows (with certain portions omitted and presented as (...)). We can observe that the readability of law corpora appears to be lower than that of the other two domains.
>
>
> ```text
> 92 P.3d 294 (2004)
> 2004 WY 71
> STATE of Wyoming, ex rel., WYOMING WORKERS' SAFETY AND COMPENSATION DIVISION, Appellant (Petitioner),
> v.
> Anthony N. SAVICKI, Appellee (Respondent).
> (...)
> We have previously entertained the Division's dissatisfaction with our failure to consider subsequent wage increases based upon merit and said this:
> (...)
> [¶ 16] Conner rejected the notion that an employee may be penalized by wage increases earned for merit at a later time. We reiterate that the pivotal question is whether the employee's injury has caused a loss of earning capacity and, if so, whether a comparison of the pre-injury wage to the post-injury wage offered is compensable under the applicable statute. In this case, Savicki's post-injury wage is compensable, and the order awarding benefits is affirmed.
> ```
>
> (continue in part 3)

---

> ### Author Response · Authors · 2023-11-21
> **Response to Reviewer aRRU (Part 3)**
>
> ### Q4: Does the approach benefit another type of models?
>
> Thank you for the insightful suggestion. As shown in the General Response above, **we add experiments on another type of model---Pythia [4]---and demonstrate positive effects of our method!**
>
> ### References:
>
> [1] Mozes van de Kar et al. Don’t prompt, search! mining- based zero-shot learning with language models. In EMNLP, Association for Computational Linguistics, 2022.
>
> [2] Leo Gao et all. The pile: An 800gb dataset of diverse text for language modeling. CoRR, abs/2101.00027, 2021.
>
> [3] Hongyang Yang et all. Fingpt: Open-source financial large language models. CoRR, abs/2306.06031, 2023.
>
> [4] Stella Biderman et al. Pythia: A suite for analyzing large language models across training and scaling. In ICML, volume 202 of Proceedings of Machine Learning Research, PMLR, 2023.

---

### Official Review · Reviewer_Cte9 · 2023-11-01

**Soundness:** 3 good
**Presentation:** 3 good
**Contribution:** 3 good
**Rating:** 6
**Confidence:** 3

**Summary:**

This paper introduces an approach for continuing the pretraining of Large Language Models (LLMs) on domain-specific corpora. Initially, the authors find that conventional domain-adaptive pretraining enhances knowledge probing while detrimentally affecting the model's prompting ability. Subsequently, they propose transforming domain-specific documents into a reading-comprehension style format. In this format, certain sentences are altered via mining patterns to pose NLP tasks such as summarization and commonsense reasoning, accompanied by their respective answers. The experiments, spanning the biomedical, finance, and law domains, demonstrate that the proposed approach yields marginal improvements in the performance of LLMs on domain-specific tasks.

**Strengths:**

- The proposed approach is straightforward yet effective in enhancing performance on domain-specific tasks, which has potential applicability to other models and domains.
- The experiments include three representative domains and six mining tasks, which may have sufficient coverage in terms of domains and tasks.
- The paper is well-written and easy to follow.

**Weaknesses:**

- In Table 4, the performance improvements appear marginal to me (3% in biomedicine, 4.8% in finance, and 4.3% in law). I am uncertain whether the benefits gained from using the proposed approach justify the effort required to transform corpora into reading comprehension texts.
- The authors only use the LLaMA 7B model for verifying their proposed method. It remains unclear whether the approach is similarly effective for smaller and larger models.

**Questions:**

- Texts in corpora may possess their own document structure, and mining NLP tasks could potentially disrupt it, impacting readability and coherence. Have the authors manually verified whether their transformation is effective without compromising the integrity of the text?

---

> ### Author Response · Authors · 2023-11-21
> **Response to Reviewer Cte9**
>
> Thanks for your positive attitude! We hope the following analysis can address your concerns about performance improvement, effectiveness for smaller and larger models, and integrity of transformation.
>
> ### Q1: Whether the performance improvements are significant.
> **Demonstrating an average improvement on our evaluation benchmarks is challenging** because the diversity of tasks within each domain requires consistent gains across multiple tasks.
>
> As shown below, in the biomedicine domain, MedAlpaca-7B, trained on diverse domain-specific data, exhibits significant improvements on certain tasks (e.g., 7.6% on ChemProt and 2.2% on USMLE). However, the improvements are not consistent across other tasks, resulting in an average improvement of 0.9%, whereas we achieve 3.1%. This trend is also noticeable in LLaMA-13B, where MedAlpaca-13B outperforms the LLaMA-13B by 0.8% on average, while we achieve a 5.1% absolute gain.
>
> |                                      | LLaMA-7B    |           |          | LLaMA-13B   |           |          |
> |--------------------------------------|-------------|-----------|----------|-------------|-----------|----------|
> |                                      | General LLM-7B | MedAlpaca-7B | AdaptLLM-7B | General LLM-13B | MedAlpaca13B |AdaptLLM-13B* |
> | PubMedQA | 59.6        | 58.6      | 63.3     | 59.6        | 60.7      | 62.5     |
> | ChemProt | 31.4        | 39.0      | 35.2     | 42.8        | 38.4      | 44.4     |
> | MQP      | 50.7        | 50.7      | 54.4     | 49.3        | 57.4      | 72.8     |
> | RCT      | 45.1        | 40.8      | 50.4     | 56.7        | 51.3      | 54.4     |
> | USMLE    | 34.5        | 36.7      | 33.1     | 34.7        | 39.0      | 34.6     |
> | AVERAGE                              | 44.2        | 45.1      | 47.3     | 48.6        | 49.4      | 53.7     |
> | **Absolute gain w.r.t. General LLM** | /           | +0.9      | **+3.1** | /           | +0.8      | **+5.1** |
>
> (* AdaptLLM-13B is trained from LLaMA-13B using our method)
>
> In the law domain, LexGPT outperforms its base model GPT-J on one task but does not exhibit the same trend on other tasks, resulting in less favorable average performance. In finance, our average improvement enables us to compete with BloombergGPT-50B.
>
> ### Q2: Whether the approach is similarly effective for smaller and larger models.
> Thanks for your helpful suggestion, please refer to the General Response above to see that **our approach remains effective for smaller and larger models!**
>
> ### Q3: Have the authors manually verified whether their transformation is effective without compromising the integrity of the text?
>
> **Yes! We have manually checked the transformed texts** before the training process. First, in the mining stage, our strategy is based on [1], where the authors have verified the reliability of the mining strategy. Besides, we have made some simple but effective modifications, such as requiring longer sentence lengths and avoiding the newline character `\n`, to ensure data quality. Second, in the transformation stage where the data is converted into reading comprehension, each transformation template has been manually checked.
>
> To further address your concerns, we sample 200 reading comprehension texts from each domain and **manually evaluate the readability and coherence. The results indicate high satisfaction**:
>
> |                           | BioMed. | Finance | Law    |
> |---------------------------|---------|---------|--------|
> | Quality of Transformation | 95.8  | 89.0  | 86.5 |
>
> ### Reference:
>
> [1] Mozes van de Kar et al. Don’t prompt, search! mining-based zero-shot learning with language models. In EMNLP, Association for Computational Linguistics, 2022.

---

### Author Response · Authors · 2023-11-21
**General Response to All Reviewers (Part 2)**

For your reference, the experimental settings are as follows:

- **Smaller and Different: Pythia-70M [1]:** Pythia-70M is a recently released and representative language model with only 70 million parameters. It shares the same architecture with the well-known GPT-NeoX [4] and can serve as a model with a different architecture than LLaMA.

- **Larger: LLaMA-13B [2]:** We scale up our base model to the one with 13 billion parameters. Due to computational constraints, we reduce the maximum sequence length to 512 during continued training.

- **After RLHF: LLaMA-2-Chat-7B [3]:** We apply our method to LLaMA-2-Chat-7B, which has undergone supervised fine-tuning and reinforcement learning with human feedback (RLHF) to align with human preferences.

	**LLaMA-2-Chat requires specific data formats**; the data format example provided by Hugging Face [5] is as follows:
	```text
	<s>[INST] <<SYS>>
	{{ system_prompt }}
	<</SYS>>

	{{ user_msg_1 }} [/INST] {{ model_answer_1 }} </s><s>[INST] {{ user_msg_2 }} [/INST] {{ model_answer_2 }}</s>
	```

	Surprisingly, we observe that our **reading comprehension can perfectly fit the data format** by transforming the reading comprehension into a multi-turn conversation! For example, we can transfer the reading comprehension in our paper's Figure 1 to a conversation like this:
	```text
	<s>[INST] <<SYS>>
	You are a helpful, respectful and honest assistant. Always answer as helpfully as possible, while being safe.  Your answers should not include any harmful, unethical, racist, sexist, toxic, dangerous, or illegal content. Please ensure that your responses are socially unbiased and positive in nature.

	If a question does not make any sense, or is not factually coherent, explain why instead of answering something not correct. If you don't know the answer to a question, please don't share false information.
	<</SYS>>

	Here is the first part of an article about biomedicine: Recent reported evidence indicates that vocal cord carcinoma is evolving similarly to oropharyngeal cancer with an increasing number of patients (...)
	Answer questions based on the article:
	What is a summary? [/INST] Glottic Carcinoma in Young Patients.
	</s><s>[INST]
	Generate a sentence that includes these biomedicine keywords [carcinoma, oropharyngeal, papillomavirus]: [/INST] Recent reported evidence indicates that vocal cord carcinoma is evolving…
	</s><s>[INST]
	Premise:… Hypothesis:… Does the premise entail the hypothesis? [/INST] Yes
	</s><s>[INST]
	What is the reason for "…"? [/INST] the morphology of the lesions and the patients' young age.
	</s><s>[INST]
	Compose a sentence that contradicts the meaning of "Historically, glottic carcinoma …".
	Answer: [/INST] Recent published evidence …
	</s><s>[INST]
	How would you complete the article? [/INST] This finding further supports…</s>
	```
	Here, we employ the system prompt used in the Hugging Face LLaMA-2 chat demo [5].

	Note that when training LLaMA-2-Chat, each training data input contains only one piece of single/multi-turn conversation, because we empirically find that concatenating multiple conversations as one data input would lead to uncontrollable gradient overflow, which would quickly shut down the training.

	In the evaluation stage, we adhere to this data format and transform each few-shot prompt into a multi-turn conversation.


### References:

[1] Stella Biderman et al. Pythia: A suite for analyzing large language models across training and scaling. In ICML, volume 202 of Proceedings of Machine Learning Research, PMLR, 2023.

[2] Hugo Touvron et al. Llama: Open and efficient foundation language models. CoRR, abs/2302.13971, 2023a.

[3] Hugo Touvron et al. Llama 2: Open foundation and fine-tuned chat models. CoRR, abs/2307.09288, 2023b.

[4] Sid Black et al. Gpt-neox-20b: An open-source autoregressive language model. CoRR, abs/2204.06745, 2022.

[5] Llama 2 is here - get it on Hugging Face

---

### Author Response · Authors · 2023-11-21
**General Response to All Reviewers (Part 1)**

### Many thanks for the positive feedback from all reviewers!
**First of all, we express our extreme gratitude to all reviewers for their positive feedback!** They appreciate the simplicity and effectiveness of our method, and also recognize its potential for broader applicability!


### Applicability to other models
In our paper, we use the 7B LLaMA—--a representative and commonly used language model—--as the base model for adaptation, and the reviewers are curious about whether our method is similarly effective for other language models i.e., smaller or larger models (by Reviewer Cte9 and Reviewer pD3t), models of different architectures (by Reviewer aRRU), or the models that have undergone RLHF (by Reviewer pD3t). Thanks for their insightful suggestions, we add experiments across various language models: **Smaller and Different: Pythia-70M [1]**, **Larger: LLaMA-13B [2]**, **After RLHF: LLaMA-2-Chat-7B [3]**.

The experiment results shown below demonstrate **consistent improvements of our methods across various models** in the biomedicine domain. We are currently working hard on expanding the experiments to other domains, and we will make all the adapted models publicly available to contribute to future research! We agree with the reviewers that the effectiveness across models is very important, so we have included this part in our paper as well.

|          | **Pythia-70M**   |          | **LLaMA-13B**   |          | **LLaMA-2-Chat-7B** |          |
|----------|-------------|----------|-------------|----------|--------------------|----------|
|          | General LLM | AdaptLLM | General LLM | AdaptLLM |     General LLM    | AdaptLLM |
| PubMedQA | 34.4        | **50.3** |    59.6     | **62.5** |        55.2        | **62.4** |
| ChemProt | 7.6         | **10.4** |    42.8     | **44.4** |        33.8        | **36.8** |
|    MQP   | **52.1**    |   50.5   |    49.3     | **72.8** |        59.3        | **63.9** |
|    RCT   | 29.6        | **30.6** |   **56.7**  |   54.4   |        53.4        | **58.2** |
|   USMLE  | 24.0        | **24.4** |   **34.7**  |   34.6   |        29.1        | **34.6** |
|  AVERAGE | 29.5        | **33.2** |    48.6     | **53.7** |        46.2        | **51.2** |

(continue in part 2)

---

### Meta-Review · Area_Chair_kcY9 · 2023-11-27

**Metareview:**

This paper introduces an approach to continue the pretraining of Large Language Models (LLMs) on domain-specific corpora for domain-adaptation. The work is motivated by authors' observation that conventional domain-adaptive pretraining enhances knowledge probing while hurts LLMs' prompting ability. Thus, the paper proposes to transform domain-specific documents into a reading-comprehension style format and then conduct continual pretraining of LLMs on the altered corpora. In the transformation, authors mined patterns from raw text in LLM corpora to build NLP tasks such as summarization and commonsense reasoning, accompanied by their respective answers. The emprirical results on three (biomedicine, law, and finance) domain-specific QA tasks suggest the effectiveness of the proposed method on llama-7b (original submission) and 13b (during rebuttal) models.

Strength:
 - The paper works on a popular topic of LLMs training and proposes techniques to improve LLM domain-specific performance without hurting general prompting capabilities. The results are of broad interests to ICLR and machine learning community.
 - The paper is well written and the solution seems simple and has broad applicability to other tasks/domains.
 - The paper conducts detailed experiments and ablation study to support the effectiveness of proposed method, sometimes applying proposed approach to 7B models can outperform much larger models (50B).
 - The work is motivated by an interesting observation: Pre-training on domain-specific datasets leads to a decline in LLM's prompting ability. This observation might open doors to more research topics.

Weakness:
 - The idea of reformating pretraining raw data into other format, such as reading comprehension or other tasks, is not new and has been proved beneficial with BERT etc. foundation models.

**Justification For Why Not Higher Score:**

The performance gain is not ground breaking and the core methodology is not entirely new. Also the approach only verified on llama 7b/13b models. Thus, I would love to see more consistent improvement across various LLMs, sizes, tasks, etc. before putting this work at spotlight.

**Justification For Why Not Lower Score:**

The strength of the paper listed above (working on a popular topics and achieving non-trivial improvement)  justify a broad interests to ICLR community and the acceptance of this work.

---

### Decision · Program_Chairs · 2024-01-16

Accept (poster)